# Identifying *Plasmodium falciparum* transmission patterns through parasite prevalence and entomological inoculation rate

Benjamin Amoah[1‡]*, Robert S McCann[2,3,4‡], Alinune N Kabaghe[3,5], Monicah Mburu[2,3], Michael G Chipeta[3,6,7], Paula Moraga[1,8], Steven Gowelo[2,3], Tinashe Tizifa[3,5], Henk van den Berg[2], Themba Mzilahowa[3], Willem Takken[2], Michele van Vugt[5], Kamija S Phiri[3], Peter J Diggle[1], Dianne J Terlouw[3,6,9†], Emanuele Giorgi[1†]

[1]Centre for Health Informatics, Computing, and Statistics (CHICAS), Lancaster Medical School, Lancaster University, Lancaster, United Kingdom; [2]Laboratory of Entomology, Wageningen University and Research, Wageningen, Netherlands; [3]Department of Public Health, College of Medicine, University of Malawi, Blantyre, Malawi; [4]Center for Vaccine Development and Global Health, University of Maryland School of Medicine, Baltimore, United States; [5]Academic Medical Centre, University of Amsterdam, Amsterdam, Netherlands; [6]Malawi-Liverpool Wellcome Trust Research Programme, Blantyre, Malawi; [7]Big Data Institute, University of Oxford, Oxford, United Kingdom; [8]Computer, Electrical and Mathematical Sciences and Engineering Division, King Abdullah University of Science and Technology (KAUST), Thuwal, Saudi Arabia; [9]Liverpool School of Tropical Medicine, Liverpool, United Kingdom

*For correspondence: mcbengy39@yahoo.com

[†]These authors contributed equally to this work
[‡]These authors also contributed equally to this work

**Competing interests:** The authors declare that no competing interests exist.

## Abstract

**Background:** Monitoring malaria transmission is a critical component of efforts to achieve targets for elimination and eradication. Two commonly monitored metrics of transmission intensity are parasite prevalence (PR) and the entomological inoculation rate (EIR). Comparing the spatial and temporal variations in the PR and EIR of a given geographical region and modelling the relationship between the two metrics may provide a fuller picture of the malaria epidemiology of the region to inform control activities.

**Methods:** Using geostatistical methods, we compare the spatial and temporal patterns of *Plasmodium falciparum* EIR and PR using data collected over 38 months in a rural area of Malawi. We then quantify the relationship between EIR and PR by using empirical and mechanistic statistical models.

**Results:** Hotspots identified through the EIR and PR partly overlapped during high transmission seasons but not during low transmission seasons. The estimated relationship showed a 1-month delayed effect of EIR on PR such that at lower levels of EIR, increases in EIR are associated with rapid rise in PR, whereas at higher levels of EIR, changes in EIR do not translate into notable changes in PR.

**Conclusions:** Our study emphasises the need for integrated malaria control strategies that combine vector and human host managements monitored by both entomological and parasitaemia indices.

**Funding:** This work was supported by Stichting Dioraphte grant number 13050800.

## Introduction

National malaria control programmes, working in collaboration with global stakeholders, have achieved extensive intervention coverage over the last two decades, leading to significant reductions in morbidity and mortality due to malaria (*Bhatt et al., 2015a*). However, malaria is still a leading global health problem. The previous successes and current challenges have motivated ambitious, yet feasible, global and national targets towards malaria elimination. A key component of efforts to achieve these targets is surveillance and monitoring, which is critical for continued assessment of intervention effectiveness, identification of areas or groups at the highest risk, and guiding the development and implementation of new intervention strategies (*World Health Organization, 2015*).

A wide range of metrics exists for monitoring malaria parasite transmission. The strengths and limitations of each metric are related, in part, to the step of the parasite transmission cycle it measures (*Tusting et al., 2014*). These strengths and weaknesses, including the sensitivity of each metric, vary across epidemiological settings and as parasite transmission declines within a given setting (*The malERA Refresh Consultative Panel on Characterising the Reservoir and Measuring Transmission, 2017*). Two of the most commonly monitored metrics are the prevalence of *Plasmodium* parasites and the entomological inoculation rate (EIR), especially in moderate to high transmission settings.

The prevalence of *Plasmodium* parasites in the human population at a given time point (i.e. the parasite rate; PR) approximates the reservoir of hosts potentially available to transmit the parasite from humans to mosquitoes. While only the gametocyte stage of the parasite contributes to transmission, it remains relatively expensive to detect this stage of the parasite. Conversely, rapid diagnostic tests (RDTs) primarily detect asexual-stage antigens, yet they are inexpensive and easily deployed in large-scale community-based surveys (*Poti et al., 2020*). Still, the limit of detection (50–200 parasites/l) for RDTs is higher than that of expert microscopy or PCR (*Chiodini, 2014*), so that RDT-based estimates of PR are biased by excluding low-density infections. Despite these limitations, RDT-based cross-sectional surveys to measure PR capture both symptomatic and asymptomatic infections, which is important because both are likely to contribute to transmission (*Bousema et al., 2014*; *Slater et al., 2019*), and changes in PR over time can indicate changes in transmission.

EIR provides an estimate of the intensity of parasite transmission from mosquitoes to humans, expressed as the number of infectious bites received per person per unit time. EIR is calculated by multiplying the number of malaria vector bites per person per unit time, also known as the human biting rate (HBR), by the proportion of vectors carrying the infectious sporozoite stage of malaria parasites, referred to as the sporozoite rate (SR) (*Onori and Grab, 1980*). The accuracy and precision of EIR estimates, therefore, depends on the accuracy and precision with which HBR and SR can be measured (*Tusting et al., 2014*). Two common methods for measuring HBR are the human landing catch and the Centers for Disease Control and Prevention Light Trap, but inter-individual variation in attractiveness to mosquitoes restricts standardisation across sampling points for both of these methods (*Knols et al., 1995*; *Qiu et al., 2006*). Alternative methods for estimating HBR include the Suna trap, which uses a synthetic blend of volatiles found on human skin and carbon dioxide to attract host-seeking *Anopheles* mosquitoes (*Mukabana et al., 2012*; *Menger et al., 2014*; *Hiscox et al., 2014*). The standardised odour blend allows for reliable comparisons among trapping locations (*Mburu et al., 2019*). Regardless of the method used to estimate HBR, the precision of SR decreases as the number of mosquitoes collected decreases. Despite these limitations, EIR is a vital metric of malaria parasite transmission because it directly describes human exposure to malaria parasites before post-inoculation factors such as immunity, nutrition, and access to health care (*Killeen et al., 2000*). Moreover, EIR provides information about the relative contributions of different vector species to transmission, which can impact malaria intervention effectiveness based on interspecies differences in biting behaviours related to time and location, non-human blood-meal hosts, larval ecology, and insecticide resistance profiles (*Ferguson et al., 2010*).

Malaria parasite transmission is heterogeneous in space and time at fine resolution due to several factors, including the availability of larval mosquito habitat, socioeconomics, human behaviour and genetics, and malaria intervention coverage (*Carter et al., 2000*; *Bousema et al., 2012*; *McCann et al., 2017a*). Repeated cross-sectional surveys continuously carried out in communities can reveal this fine-resolution heterogeneity (*Roca-Feltrer et al., 2012*), providing timely estimates

of malaria control progress at the sub-district level and potentially identifying hotspots of malaria parasite transmission for targeted intervention (*Kabaghe et al., 2017*; *Bousema et al., 2016*). However, understanding this heterogeneity and identifying hotspots in a way that is meaningful for control programmes remains challenging (*Stresman et al., 2019*), in part because hotspot location and size can depend on which metric is used (*Stresman et al., 2017*). Given that PR and EIR are indicative of components of the parasite transmission cycle that are separated by multiple complex steps, each metric provides partial but useful information about the underlying risk of transmission. Therefore, measuring and mapping both metrics can provide a fuller picture of parasite transmission (*Cohen et al., 2017*).

Additionally, modelling the functional relationship between EIR and PR can provide further insights into the underlying malaria epidemiology. Previous studies have demonstrated that this relationship is non-linear, such that small changes in EIR are associated with large changes in PR when EIR is low, but PR saturates rather than changing at a constant rate when EIR is high (*Beier et al., 1999*; *Smith et al., 2005*). These previous studies were meta-analyses using paired estimates of EIR and PR, with one estimate of each outcome per site, from sites representing a wide range of EIR and PR in Africa. Their findings had a number of important implications, which included providing estimated ranges for the change in PR that may be expected for a given change in EIR. However, these estimates implicitly assumed that the relationship is constant across space on a continental scale, such that differences in EIR and PR between sites would be indicative of differences over time within a site. Yet no previous study has explicitly examined this relationship over time within a single geographical region.

In the current study, we use a series of repeated cross-sectional surveys conducted over 38 months in one region of southern Malawi to map the fine-scale spatiotemporal dynamics of *P. falciparum* entomological inoculation rate (PfEIR) and *P. falciparum* parasite prevalence (PfPR). The joint monitoring of these two outcomes in space and time allows us to identify and compare the spatial heterogeneities and temporal patterns of PfEIR and PfPR in a region with moderately intense, seasonally variable malaria parasite transmission. We then investigate the PfEIR-PfPR relationship based on changes in these outcomes observed at both annual and subannual scales within our study site using several statistical models, which can be distinguished as follows: mechanistic models that are based on different epidemiological assumptions and empirical models where the data inform the PfEIR-PfPR relationship. These approaches allows us to address the following questions. (1) How do spatiotemporal patterns of EIR and PR compare? (2) Do EIR and PR lead to the identification of the same malaria hotspots? (3) As EIR changes over time, how do those changes in EIR affect PR? (4) Does EIR have a lagged effect on PR? (5) Does the EIR-PR relationship vary between women of reproductive age and children between 6 and 59 months of age?

## Materials and methods

### Study site

This study was part of the Majete Malaria Project (MMP), an integrated malaria control project in Chikwawa District, Malawi. The catchment area of MMP consisted of three distinct geographical regions, referred to as Focal Areas A, B, and C (*Figure 1*), with a total population of about 25,000 people living in 6600 households in 65 villages.

Chikwawa experiences highly variable rainfall during its single rainy season, which spans November/December to April/May. Temperatures are generally high, with daily maximum temperatures in December averaging 37.6°C, and in July averaging 27.6°C (*Joshua et al., 2016*). A wide range of both permanent and temporary water bodies create suitable larval habitats in the region for *Anopheles funestus s.s.*, *Anopheles arabiensis*, and *Anopheles gambiae s.s.*, including dams, swamps, ponds, borehole runoffs and drainage channels (*Gowelo et al., 2020*).

Malaria control in the district is implemented through the Chikwawa District Health Office. During the study period, interventions applied throughout the study area included the continuous provision of insecticide-treated nets (ITNs) to pregnant women and children under five years old, mass distribution campaigns of ITNs targeting universal coverage, intermittent preventative therapy for pregnant women, and malaria case diagnosis and treatment with artemisinin-based combination therapy. The only mass distribution of ITNs in the district during the study period occurred in April 2016. As

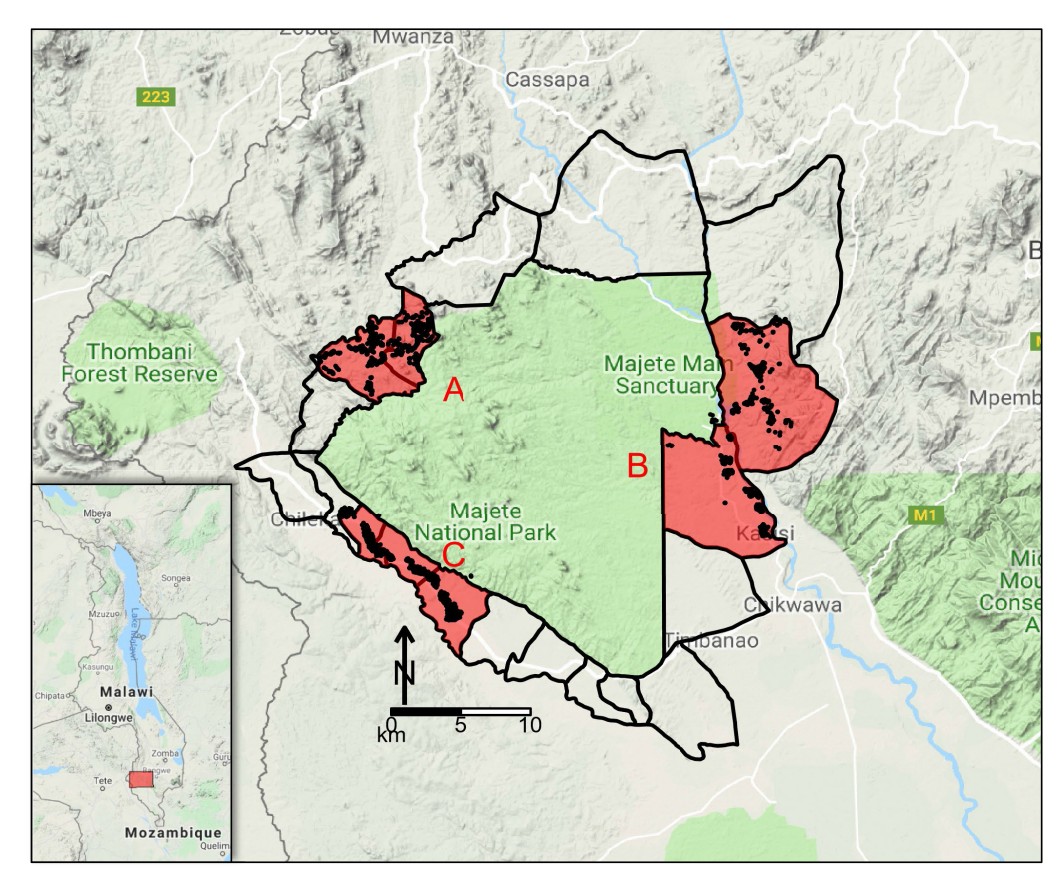

**Figure 1.** Map of study site. Map of Malawi (insert) highlighting the Majete Wildlife Reserve and the borders of 19 community-based organisations (CBOs) surrounding the Majete perimeter. Three focal areas (red patches), labelled as A, B, and C, show the households (black points) selected for the parasitaemia and entomological surveys by the Majete Malaria Project (MMP). The base map was obtained from Google Maps.

part of the MMP, a randomised trial was conducted to assess the effectiveness of additional, community-implemented malaria interventions between May 2016 and May 2018 (*McCann et al., 2017b*). The trial interventions were implemented at the village level, with villages assigned to one of four groups: (a) no additional interventions; (b) larval source management; (c) house improvement; and (d) both larval source management and house improvement (*McCann et al., 2017b*; *van den Berg et al., 2018*).

## Data

To quantify PfPR and PfEIR over the course of the study, a rolling malaria indicator survey (rMIS) (*Roca-Feltrer et al., 2012*) was conducted in conjunction with mosquito sampling, forming a series of repeated cross sectional surveys. Sampling was carried out over 17 rounds, with each round spanning a period of 2 or 3 months. In the first two rounds of data collection (baseline, from April through August 2015), an inhibitory geostatistical sampling design (IGSD) was used to select 300 and 270 households, respectively, for the rMIS from an enumeration database of all households in the catchment area (*Chipeta et al., 2017*). The IGSD helped to ensure that randomly sampled households are relatively uniformly spaced over the study region by requiring each pair of sampled households to be separated by a distance of at least 0.1 km, which increases the efficiency of hot-spot detection (*Kabaghe et al., 2017*). In the three subsequent rounds of data collection during the baseline, an adaptive geostatistical sampling design (AGSD) was used to select 270 households per round (*Chipeta et al., 2016*). With AGSD, new households for the current round of rMIS were chosen from regions with high standard errors of estimated prevalence, based on data from all previous rounds. In the baseline period, previously sampled households were not eligible for sampling in

subsequent rounds. For the trial period (starting May 2016), IGSD was again used to select households from the enumeration database of all households. All households were eligible for selection in each round of the trial period regardless of whether they were selected in a previous round. At each round of rMIS data collection in the baseline and trial phases, respectively, 75% and 72% of the households chosen at each round of the rMIS were then randomly selected for mosquito sampling.

In each sampled household, children under five (0.5–5 y/o) and women of reproductive age (15–49 y/o) were tested for *P. falciparum* using an RDT (SD BIOLINE Malaria Ag P.f. HRP-II, Standard Diagnostics, Yongin-si, Republic of Korea).

Mosquitoes were sampled from 5pm to 7am using Suna traps (Biogents AG, Regensburg, Germany) with MB5 blend plus $CO_2$ to mimic human odour (*Hiscox et al., 2014*; *Mburu et al., 2019*). For a selected household in a surveillance round, the trap was set for one night indoors and one night outdoors, with the order of indoor/outdoor determined randomly. For households where the residents were sleeping in more than one building, a trap was set at each building. Trapped female anophelines were preserved using a desiccant and identified using standard morphological and molecular techniques (*Gillies and Coetzee, 1987*; *Koekemoer et al., 2002*; *Scott et al., 1993*). Female anophelines were further tested for the presence of *P. falciparum* in their head and thorax, after removing the abdomen, using quantitative polymerase chain reaction (qPCR) (*Bass et al., 2008*; *Perandin et al., 2004*). Specimens with a Ct value below 37.0 were considered positive for *P. falciparum*.

## Environmental and climatic factors

Environmental and climatic factors affect the abundance and suitability of water bodies that support the development of immature mosquitoes (*Madder et al., 1983*; *Loetti et al., 2011*), the duration of mosquito development (*Ciota et al., 2014*; *Loetti et al., 2011*; *Craig et al., 1999*), mosquito host-seeking and biting behaviour, and the development rate of malaria parasites in mosquitoes (*Rumisha et al., 2014*; *Amek et al., 2011*).

Using hourly measurements of temperature and relative humidity (RH) from a weather station in each focal area, we computed the average temperature and RH for different ranges of days before the day of data collection (Appendix 1 – Procedure for building the HBR, PfSR and PfPR models).

Spectral indices, namely normalised difference vegetation index (NDVI) and enhanced vegetation index (EVI), were computed using remotely sensed multi-spectral imagery from the Landsat 8 satellite. These data are freely available from the United States Geological Survey (USGS) Earth Explorer (earthexplorer.usgs.gov) as raster files at a spatial resolution of 30 × 30 m for every 16 days. For our analysis, we averaged each spectral index over 5 years, from April 2013 to April 2018, while omitting scenes that were dominated by cloud artefacts.

We extracted raster data of surface elevation from the global digital elevation model (DEM) generated using measurements from the Advanced Space-borne Thermal Emission and Reflection Radiometer (ASTER) (*Tachikawa et al., 2011*). These data are also freely available for download from the USGS Earth Explorer. Using a flow accumulation map derived from the DEM, a river network map was generated and used to calculate and store as raster images the distance to small rivers and large rivers (henceforth, DSR and DLR, respectively).

## Geostatistical analysis

The number of mosquitoes trapped by Suna traps can be used to estimate HBR, as these traps primarily target host-seeking mosquitoes. Hence, we first estimated HBR and the *P. falciparum* sporozoite rate (PfSR). We then estimated PfEIR as the product of these two quantities.

We carried out separate analyses for *A. arabiensis* and *A. funestus s.s.*, using explanatory variables and random effects structures that we found to be suitable for each species. Details of the variable selection process and the final sets of explanatory variables for each of the models later described in this section are given in Appendix 1 – Procedure for building the HBR, PfSR, and PfPR models. The correlation structures adopted for the geostatistical models were informed by the variogram-based algorithm described in *Giorgi et al., 2018*. The geostatistical models for the HBR and PfPR data described below were fitted using PrevMap (*Giorgi and Diggle, 2017*), freely available from the Comprehensive R Archive Network (CRAN, www.r-project.org). The PfSR models were fitted using the glmm package, also available on CRAN.

## Human biting rate

Let $Y(x_i, t_i)$, $i = 1, \ldots, M$, where $M = 2432$ is the total number of houses, denote counts of mosquitoes trapped at location $x_i$ in month $t_i \in \{1, \ldots, 38\}$, where $t_i = 1$ denotes April 2015. We modelled the $Y(x_i, t_i)$ using Poisson mixed models expressed by the following linear predictor

$$\log\{HBR(x_i, t_i)\} = d(x_i, t_i)^\top \beta + f(t_i; \alpha) + S(x_i) + Z_i, \tag{1}$$

where: $d(x_i, t_i)$ is a vector of spatiotemporal explanatory variables with associated regression coefficients $\beta$; the $f(t_i; \alpha)$ is a linear combination of several functions of time, including sines, cosines and splines, with an associated vector of regression parameters $\alpha$, accounting for trends and seasonal patterns; the $Z_i$ are independent and identically distributed Gaussian random variables with variance $\tau^2$; $S(x)$ is a zero-mean stationary and isotropic Gaussian process with variance $\sigma^2$ and exponential correlation function $\rho(u) = \exp(-u/\phi)$, where $\phi$ regulates the pace at which the spatial correlation decays for increasing distance $u$ between any two locations. We allow the explanatory variables $d(x_i, t_i)$ and $f(t_i; \alpha)$ to differ between different mosquito species since different species may respond differently to environmental changes. We point out that the stationarity of the process $S(x)$ implies that all of its properties, including the variance ($\sigma^2$) and scale of the spatial correlation ($\phi$), are constant over space. The estimation of the model parameters is then carried out using Monte Carlo Maximum Likelihood (*Christensen, 2004*).

## *Plasmodium falciparum* sporozoite rate

Let $Y^*(x_i, t_i)$ be the number of mosquitoes that tested positive for the presence of *P. falciparum* sporozoites. We assumed that the $Y^*(x_i, t_i)$ follow a Binomial mixed model with number of trials $N^*(x_i, t_i)$, that is the total number of successfully tested mosquitoes, and probability of testing positive $PfSR(x_i, t_i)$. We model the latter as a logit-linear regression given by

$$\log\left\{\frac{PfSR(x_i, t_i)}{1 - PfSR(x_i, t_i)}\right\} = d(x_i, t_i)^\top \beta^* + f^*(t_i; \alpha^*) + Z_i^*, \tag{2}$$

where each term in (2) has an analogous interpretation to those of (1). A spatial process $S(x)$ was not included in the sporozoite rate model because we found no evidence of residual spatial correlation in the sporozoite rate data (*Appendix 1—figure 1*).

## Estimating the *Plasmodium falciparum* entomological inoculation rate

Let $PfEIR_f(x, t)$ and $PfEIR_a(x, t)$ denote the PfEIR for *A. funestus s.s.* and *A. arabiensis* at a given location $x$ and month $t$. We estimated each of these two as

$$PfEIR_f(x, t) = HBR_f(x, t)PfSR_f(x, t)l(t)$$
$$PfEIR_a(x, t) = HBR_a(x, t)PfSR_a(x, t)l(t),$$

where $l(t)$ is the number of days in month $t$. Finally, we estimated the overall PfEIR as

$$PfEIR(x, t) = PfEIR_f(x, t) + PfEIR_a(x, t). \tag{3}$$

We then mapped PfEIR as in (3) over a 30 × 30 m regular grid covering the whole of the study area for each month across 38 months.

To map PfEIR for each month, we first simulate at each prediction location (i.e. the centroid of each grid cell) 10,000 samples from the conditional distribution of the random effects (corresponding to $S(x) + Z$ in the case of the PfHBR and $Z$ in the case of PfSR) given the data. We then transform these to obtain 10,000 predicted surfaces for PfHBR and PfSR, and by applying (3) we obtain 10,000 predictive samples for PfEIR. The predicted PfEIR at each prediction location is taken to be the median of the 10,000 samples at that location. The associated 95% predictive interval is the 2.5th to 97.5th percentile of the 10,000 predictive samples.

In this procedure, all the parameters corresponding to the regression coefficients, the scale and variance of the spatial process, and variance of Gaussian noise are fixed at their MCML estimates.

### *Plasmodium falciparum* prevalence

We mapped PfPR in women and in children by fitting a geostatistical model to each group. More specifically, let $I(x_i, t_i)$ denote the number of RDT positives out of $N_{it}$ sampled individuals at location $x_i$ in month $t_i$. We then assumed that the $I(x_i, t_i)$ follow a Binomial mixed model with probability of a positive RDT result $p(x_i, t_i)$, such that

$$\log\left\{\frac{p(x_i, t_i)}{1 - p(x_i, t_i)}\right\} = d(x_i, t_i)^\top \varphi + g(t_i; \varrho) + T(x_i) + U_i, \tag{4}$$

where $T(x_i)$ is a stationary and isotropic Gaussian process with exponential correlation function and $U_i$ are Gaussian noise, $g(t_i, \varrho)$ is a linear combination of splines, and sine and cosine functions of time accounting for trends and seasonality, and $\varphi$ and $\varrho$ are vectors of regression parameters to be estimated.

### Hotspot detection using PfEIR and PfPR

We demarcated hotspots for PfEIR and PfPR using an exceedance probability approach. Using the resulting 10,000 predictive samples for PfEIR and PfPR, as described above, we then obtained the exceedance probability for each outcome at each space-time location by computing the proportion of the 10,000 predictive samples that exceeded the respective, predefined thresholds, which were set at 0.1 ib/person/month for EIR, 31% for PfPR in children, and 17% for PfPR in women. Finally, we mapped these exceedance probabilities and demarcated hotspots as areas where these probabilities were at least 0.9. The PfPR thresholds were defined to correspond to the PfEIR threshold based on the best of six functional relationships between PfEIR and PfPR as described in the next section.

## Modelling the relationship between PfEIR and PfPR

Because PfEIR may have a delayed effect on PfPR, possibly due to the time taken for *P. falciparum* to develop in the human host, we considered that current PfPR may depend on PfEIR $l$ months prior. In particular, we considered $l = 0, 1, 2$. We then assumed that the number of RDT positive individuals, $I(x_i, t_i)$, follow independent Binomial distributions such that

$$PfPR(x_i, t_i) = h\{Pf\hat{E}IR(x_i, t_i - l)\}, \tag{5}$$

where $h(\cdot)$ is a function depending on a vector of parameters $\theta$ that governs the relationship between PfPR and PfEIR, and $Pf\hat{E}IR(x_i, t_i - l)$ is the estimated PfEIR as in *Equation (3)*. We considered six models, each of which provided a different specification for $h(\cdot)$.

   We now describe the six models for $h(\cdot)$. Models 1 to 4 make explicit assumptions on the underlying mechanism of transmission, whereas models 5 and 6 describe the functional relationship between PfEIR and PfPR through regression methods.

### Model 1: The susceptible-infected-susceptible (SIS) model

Let $b$ be the probability that an infectious mosquito bite results in an infection, referred to as the transmission efficiency. Then, infections at $(x_i, t_i - l)$ are assessed to occur at a rate of $b \times PfEIR(x_i, t_i - l)$. We assumed that each infection cleared independently over a duration $1/r$ so that the ratio $\gamma = b/r$ is the time taken to clear infection per infectious bite. We assumed that the relationship between PfEIR and PfPR holds throughout the study region. If $PfEIR(x, t - l)$ is constant, the relationship between $PfEIR(x, t - l)$ and $PfPR(x, t)$ is described by *Ross, 1911*

$$\frac{\partial PfPR(x, t)}{\partial t} = b \times PfEIR(x, t - l)(1 - PfPR(x, t)) - r \times PfPR(x, t). \tag{6}$$

We obtained our first model as the non-zero equilibrium solution of (6), given by

$$PfPR(x, t) = \frac{\gamma PfEIR(x, t - l)}{\gamma PfEIR(x, t - l) + 1}. \tag{7}$$

## Model 2: The SIS model with different infection/recovery rates (D.I/R)

Model 1 assumes that women and children get infected and recover at the same rate. However, the transmission and recovery rates in children may differ from those in women. We, therefore, modified Model 1 by allowing different values of $b$ and $r$ for each category of people. Let $\xi_{1,it}$ and $\xi_{2,it}$ respectively be the proportion of children and women sampled at $(x_i, t_i)$ and $\gamma_k = b_k/r_k$, where $k = 1$ denotes children and $k = 2$ denotes women. The resulting Model 2 is

$$PfPR(x,t) = \sum_{k=1}^{2} \xi_{k,it} \frac{\gamma_k PfEIR(x,t-l)}{\gamma_k PfEIR(x,t-l)+1}. \tag{8}$$

## Model 3: The SIS model with superinfection (S.I.)

If individuals are super-infected with *P. falciparum*, then the rate at which infections clear depends on the infection rate, with clearance being faster when infection rate is low, and slower when infection rate is high. To capture this feature, we modelled infection clearance rate as $g(\vartheta, r) = \vartheta/(e^{\vartheta/r} - 1)$, where $\vartheta = b \times PfEIR$(*Smith et al., 2005*; *Walton, 1947*; *Dietz et al., 1974*; *Aron and May, 1982*). The resulting model for $PfPR(x,t)$ is

$$PfPR(x,t) = 1 - \exp\{-\gamma PfEIR(x,t-l)\} \tag{9}$$

## Model 4: The SIS model with S.I and D.I/R

Combining the assumptions of heterogeneous infection/recovery rates, as in Model 2 and superinfection, as in Model 3, we obtain Model 4,

$$PfPR(x,t) = \sum_{k=1}^{2} \xi_{k,it}(1 - \exp\{-\gamma_k PfEIR(x,t-l)\}). \tag{10}$$

## Model 5: The Beier model

*Beier et al., 1999* assumed that the log of PfEIR is linearly related to PfPR, and fitted the regression model

$$PfPR(x,t) = a + b\log(PfEIR(x,t-l)), \tag{11}$$

the so called 'log-linear model'.

## Model 6: The logit-linear model

The Beier model has the limitation that PfPR approaches $-\infty$ as PfEIR goes to 0 and approaches $\infty$ as PfEIR goes to $\infty$. To constrain PfPR to lie between 0 and 1, we applied the logit-link function to PfPR to give Model 6,

$$\log\left(\frac{PfPR(x,t)}{1 - PfPR(x,t)}\right) = a + b\log(PfEIR(x,t-l)). \tag{12}$$

## Parameter estimation of the PfEIR-PfPR relationship models

We estimated the parameters of each of the six models by maximising the log-likelihood function

$$\sum_{t_i}\sum_{x_i} I(x_i, t_i)\log(PfPR(x_i, t_i)) + (N_{it} - I(x_i, t_i))\log(1 - PfPR(x_i, t_i)). \tag{13}$$

To fit each model, we first obtained 10,000 bootstrapped data sets of predicted PfEIR as in (3) at the set of all space-time locations sampled for the rMIS. We did this for two reasons: to obtain PfEIR data at locations $(x_i, t_i)$ that were sampled for rMIS but not for the entomological surveillance; and to account for the uncertainty in PfEIR. The predicted PfEIR values were then paired with respective empirical PfPR values at $(x_i, t_i)$. By fitting each model to each of the 10,000 datasets, we then obtained 10,000 bootstrapped samples $\{\hat{\theta}_1, \ldots, \hat{\theta}_{10000}\}$ for the vector of parameter estimates $\hat{\theta}$ of each the six candidate models. We then summarised these samples by their mean and central 95% probability interval. We repeated this process for $l = 0, 1, 2$.

We compared the fit of the six models based on their predictive ability as measured by the bias and root-mean-square error when each model is used to predict prevalence at all the observed space-time locations.

## Results

### rMIS and mosquito sampling

From April 2015 to May 2018, a total of 6870 traps (3439 indoors; 3431 outdoors) were placed at 2432 houses over 17 rounds of sampling (*Figure 2*), resulting in the collection of 657 female *Anopheles* mosquitoes (*Table 1*). Following PCR of the 478 *A. gambiae* s.l. collected, 92% were identified as *A. arabiensis*, 2% as *A. gambiae s.s.*, 1% as *A. quadriannulatus*, and 5% could not be identified further. From the 179 *A. funestus* s.l. collected, 95% were identified as *A. funestus s.s.* by PCR, while the remaining 5% could not be identified further. The observed vector composition is therefore 71%, 27%, and 2% for *A. arabiensis*, *A. funestus s.s.*, and *A. gambiae s.s.*, respectively.

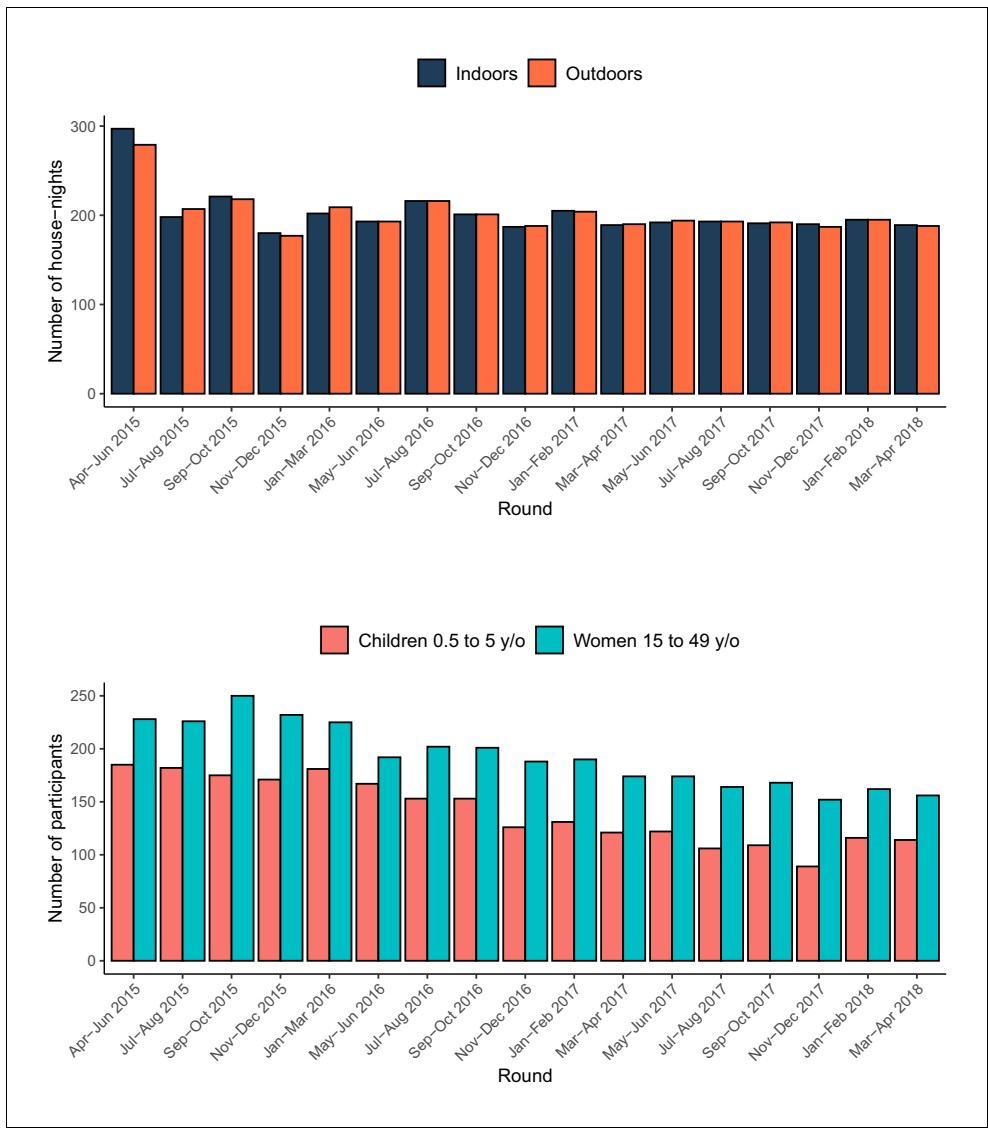

**Figure 2.** Summary of the entomological and rMIS sampling over time. The top panel shows the number of houses where Suna traps were set, and the bottom panel shows the number of participants in the rMIS.

**Table 1.** Details of *Anopheles* female mosquitoes collected.

The table shows the observed numbers collected indoors and outdoors, the HBR (number collected per trap multiplied by the number of days in each of the 38 months of sampling), PfSR and PfEIR for the *Anopheles* species sampled.

| Species | Number Collected Indoors | Number Collected Outdoors | Empirical HBR (bite/person) | Empirical PfSR % | Empirical PfEIR (ib/person) |
|---|---|---|---|---|---|
| *A. arabiensis* | 175 | 263 | 73.66 | 5.48% | 4.04 |
| *A. funestus s.s.* | 74 | 96 | 28.58 | 11.17% | 3.19 |
| *A. gambiae s.s.* | 5 | 6 | 1.85 | 18.18% | 0.34 |
| *A. quadriannulatus* | 1 | 3 | 0.67 | 0.00% | 0.00 |
| *A. gambiae s.l.** | 12 | 13 | 4.20 | 12.00% | 0.50 |
| *A. funestus s.l.*** | 4 | 5 | 1.51 | 11.11% | 0.17 |
| TOTAL | 271 | 386 | 110.47 | | 8.24 |

A. gambiae s.l. * and A. funestus s.l. ** are Anopheles female mosquitoes morphologically identified as belonging to the A. gambiae species complex and A. funestus species group, respectively, but which could not be further identified by PCR. The unit of EIR is infective bites per person over the course of the study (38 months).

The online version of this article includes the following source data for Table 1:

Source data 1. Source data of the details of *Anopheles* female mosquitoes collected.

Despite the relatively low abundance of *A. funestus s.s.* compared to *A. arabiensis*, the higher sporozoite rate of the former made the contribution of *A. funestus s.s.* to the total PfEIR almost equivalent to that of *A. arabiensis* (*Table 1*). The total PfEIR for the 38 months was 8.24 ib/person, equivalent to an average 2.60 ib/person/year.

Over the same 38-month period, 5685 individual *P. falciparum* RDT tests were conducted across 3096 household visits (*Figure 2*). Among the 2401 tests conducted on children aged 6 to 59 months, 25.5% were positive, while 14.3% of the 3284 tests conducted on women aged 15–49 y/o were positive.

## Spatiotemporal patterns of PfEIR and PfPR

We observed clear spatiotemporal heterogeneities in PfEIR, PfPR in children, and PfPR in women when mapped across the study region at a fine spatial resolution (30 x 30 m) and 1 month intervals. For convenient visualisation of the main features of the spatiotemporal maps, we have developed an interactive web-based application to show the maps at http://chicas.lancaster-university.uk/projects/malaria_in_malawi/pfpr/. We show selected predictive maps of PfEIR and PfPR in *Figure 3* and exceedance probability of PfEIR and PfPR in *Figure 4* for June 2015, August 2016 and November 2017, which are representative of high, medium and low transmission months, relative to the full study period. Spatially, there were differences both within and between the three focal areas. Focal Area A generally showed the lowest PfEIR and PfPR, while Focal Areas B and C showed similar, higher levels of PfEIR. Within each focal area, the spatial patterns changed from month to month, with hotspots of both PfEIR and PfPR proceeding through seasonal cycles of expansion and retraction over time. Over the 3-year study period, hotspots of PfEIR and PfPR generally disappeared during the low transmission seasons, except for residual hotspots of PfPR that persisted throughout the study period.

When summarised over the whole study region, each of PfPR and PfEIR exhibited seasonal patterns with a single annual peak. The monthly predicted PfEIRs and PfPRs were similar to the observed values (*Figure 5*). PfEIR increased from November to a peak in May and decreased to a trough in November. PfPR started increasing from December to a peak around July, after which it decreased to a trough between November/December.

Three observations are clear from both the spatiotemporal maps and the monthly summarised data (*Figure 5*). First, children aged 6–59 months consistently had a higher level of PfPR than women throughout the study period. Second, PfPR in both groups generally decreased from the start of the

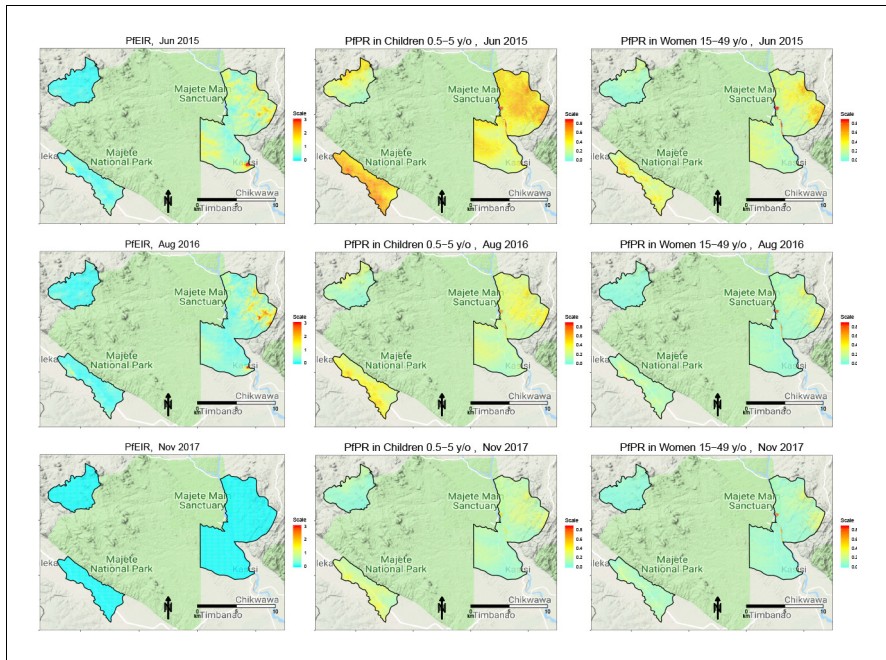

**Figure 3.** Selected predictive maps of PfEIR and PfPR for June 2015, August 2016 and November 2017, representing high, medium, and low transmission months, respectively. Left panels: median *P. falciparum* entomological inoculation rate (PfEIR). Centre panel: mean *P. falciparum* parasite prevalence (PfPR) in children 0.5–5 y/o. Right panel: mean PfPR in women 15–49 y/o.

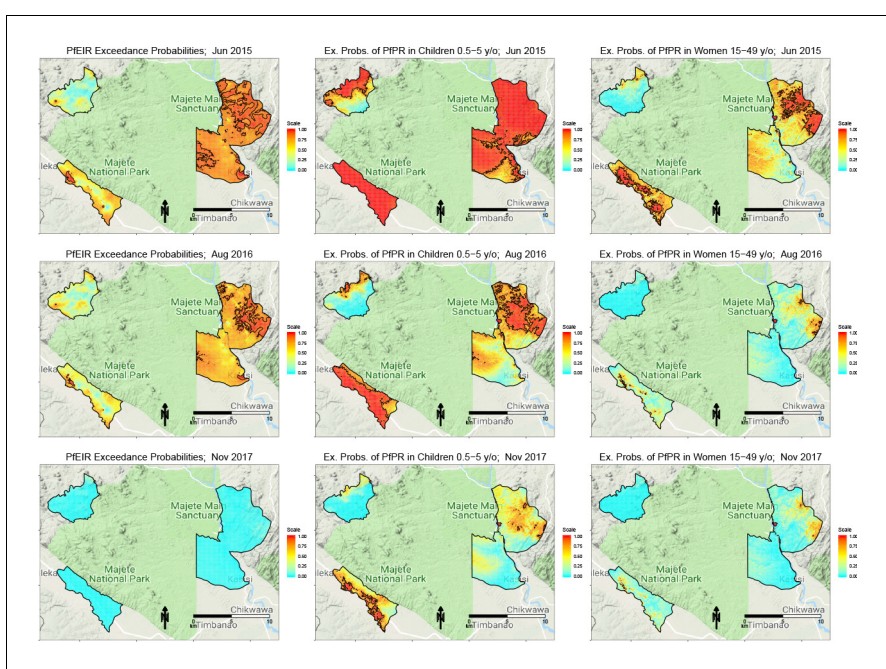

**Figure 4.** Selected maps of exceedance probability of PfEIR and PfPR for June 2015, August 2016 and November 2017, representing high, medium, and low transmission months, respectively. Left panels: probability that PfEIR exceeds 0.1 infective bites/person/month. Centre panels: probability that PfPR in children 0.5–5 y/o exceeds 31%. Right panels: probability that PfPR in women 15–49 y/o exceeds 17%. Red areas demarcate hotspots, which we define as an exceedance probability at least 0.9.

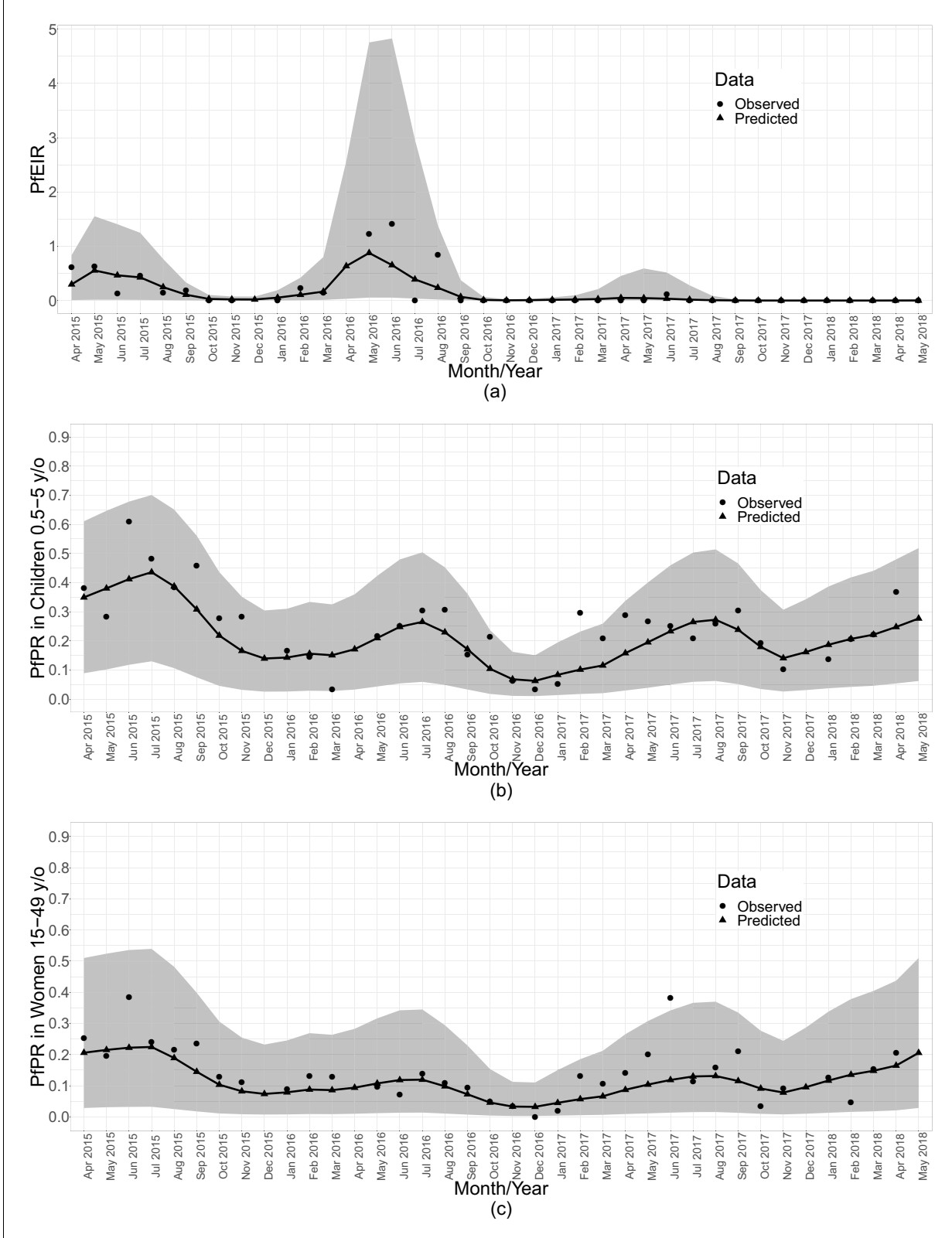

**Figure 5.** Summaries of monthly PfEIR and PfPR. The plot shows monthly median PfEIR (**a**), mean PfPR in children 0.5–5 y/o (**b**) and mean PfPR in women 15–49 y/o (**c**), over the study region. The round points are the observed data and the triangular points are the predictions from our models. The shaded regions represent the corresponding 95% confidence interval of the predicted values. The confidence intervals were obtained by simulating 10,000 samples of the respective metric under the respective fitted model.

study in April 2015 to December 2016, after which there was a general trend of increasing PfPR in both age groups. Finally, PfEIR was steady in the first 2 years of the study, followed by a general decrease after May 2016. Strikingly, the observed PfEIR was 0 between June 2017 and the end of the study, while the PfPR increased in both children and women between November 2017 and May 2018.

## The relationship between PfEIR and PfPR

Temporally, the seasonal patterns of PfEIR and PfPR within each year were nearly concurrent, with the estimated peak in PfEIR preceding that of PfPR by one month (*Figure 5*).

Spatially, PfEIR and PfPR showed broadly similar patterns. When comparing the hotspots of PfEIR and PfPR using spatiotemporal maps of exceedance probabilities, the hotspots of PfEIR and PfPR partly overlapped during the high transmission season (http://chicas.lancaster-university.uk/projects/malaria_in_malawi/pfpr/). However, there were hotspots of PfEIR that were not necessarily hotspots of PfPR and vice versa (*Figure 6*).

Scatter plots of the logit of PfPR against the log of PfEIR show an approximately direct linear relationship (*Figure 7*).

For each of the six classes of model, those with a 1-month lagged-effect were found to be the best based on the root-mean-square-error (RMSE) and bias indices of predictive performance. Among these, the empirical models (i.e. logit-linear and Beier) yielded lower values for the bias and RMSE values than the mechanistic models (Appendix 1 – Procedure for building the HBR, PfSR, and PfPR models, *Appendix 1—table 6*). The logit-linear model, albeit by a small margin, outperformed all the models (Appendix 1 – Procedure for building the HBR, PfSR, and PfPR models, *Appendix 1—table 6*).

The fitted logit-linear model (*Figure 8*) shows that PfPR rises quickly with increasing PfEIR at very low PfEIR, followed by a flattening off or saturation. From the estimated relationship for women and children combined (*Figure 8a*), a decrease in PfEIR from 1 ib/person/month to 0.85 ib/person/month (i.e. a 0.15 decrease in PfEIR) is associated with a reduction in PfPR from 27.17% to 26.85% on average (i.e. a 1.18% decrease in PfPR). However, at the lower ranges of EIR, the same decrease of 0.15 ib/person/month from 0.2 ib/person/month to 0.05 ib/person/month is associated with a reduction in PfPR from 24.10% to 21.66% on average (i.e. a 10.13% decrease in PfPR). The resulting non-linear

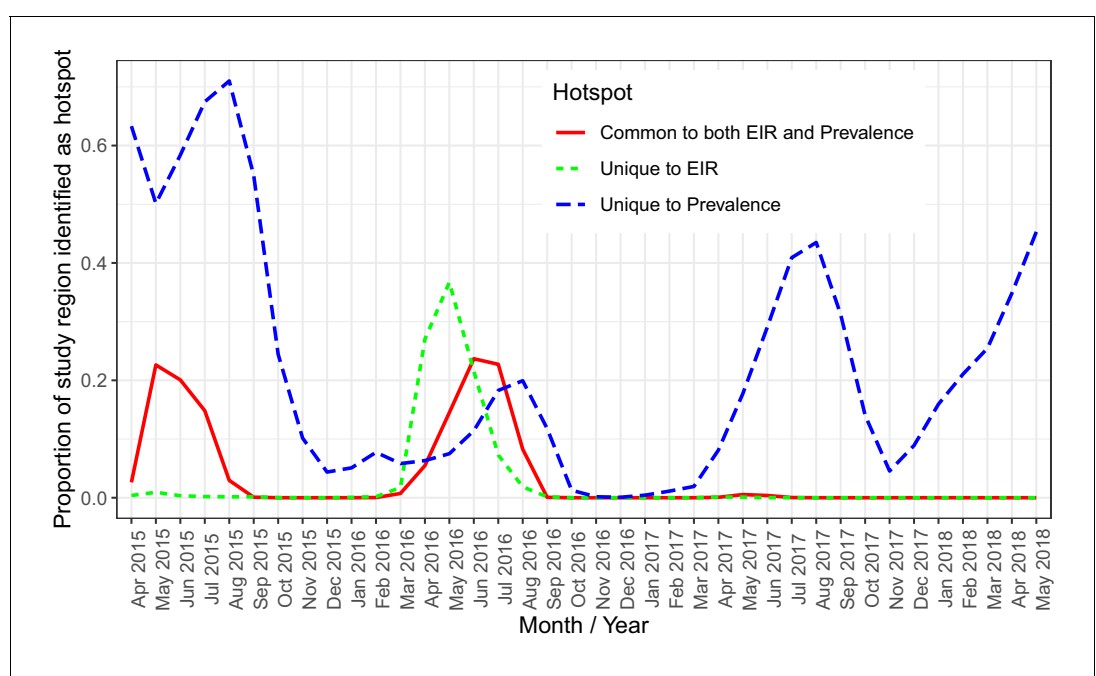

**Figure 6.** A plot of the proportion of the study region demarcated as hotspot. The solid (red) line shows hotspots identified by both PfPR and PfEIR. The long dashed (blue) line shows hotspots identified uniquely by PfPR whilst the short dashed (green) line shows hotspot uniquely identified by PfEIR.

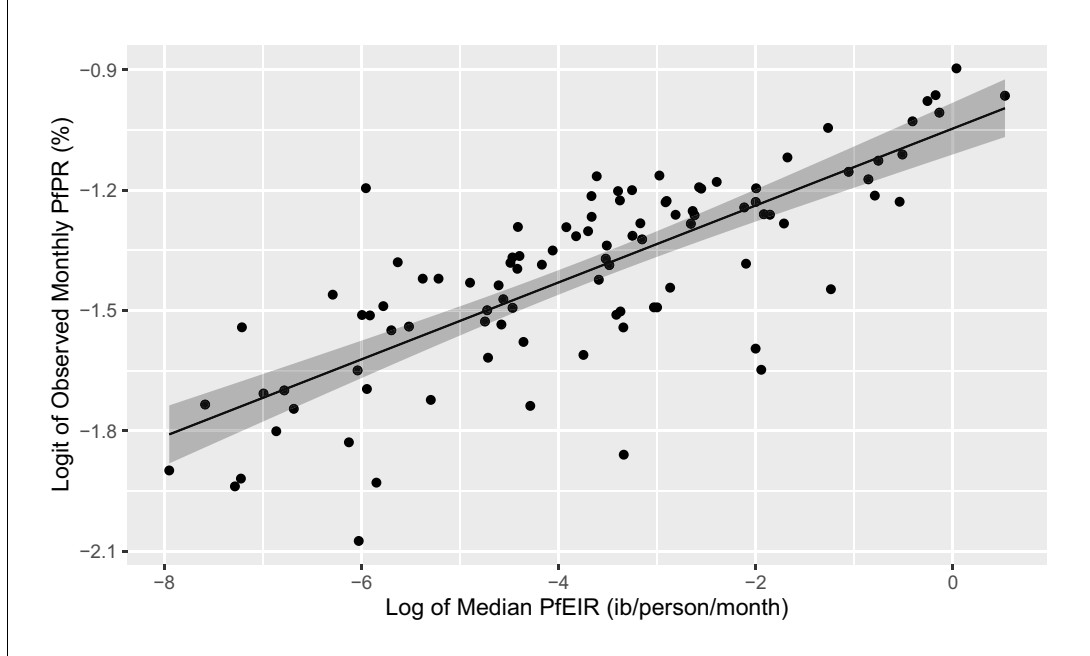

**Figure 7.** Plot of the linear relationship between the logit of PfEIR and the log of PfEIR. Each point represents a focal area and a month where there was empirical data for PfPR (n=100). PfEIR is the median (model-based predicted) PfEIR over the focal area. Prevalence is the average empirical prevalence over the focal area, with children and women put together. The shaded regions represent the corresponding 95% confidence region. The confidence region was obtain from 10,000 predictive samples where each sample was obtained by plugging in one of the bootstrap samples parameter estimate into the logit-linear model.

reltionship emphasizes two aspect of the PfPR-PfEIR relationship. First, for large values in PfEIR, reductions in PfEIR are associated with smaller reductions in PfPR, whilst if PfEIR is low, reductions in PfEIR are associated with greater reductions in PfPR. Second, even when transmission, as measured by PfEIR, has been driven close to zero, PfPR can still remain substantial.

An indication of possible differences in the PfEIR–PfPR relationship between children and women lies in the logit-linear model fitted to children and women separately (*Figure 8b*). The average trajectories of PfPR and corresponding 95% confidence intervals with varying PfEIR are distinct for women and children. PfPR in children tends to show a steeper rise with increasing PfEIR than in women. From the estimated relationship for children, a decrease in PfEIR from 0.1 ib/person/month to 0.01 ib/person/month is associated with a reduction in PfPR from 31.07% to 25.52% on average (i.e. a 17.86% decrease in PfPR). From the estimated relationship for women, the same decrease in PfEIR is associated with a reduction in PfPR from 16.84% to 14.33% (i.e. a 14.90% decrease in PfPR) on average. We make two observations. (1) With decreasing PfEIR, the percentage reduction in PfPR achieved in children tends to be higher than in women. (2) When transmission has been driven almost to zero, PfPR remains consistently high in children.

## Discussion

Using data from 38 months of repeated cross-sectional surveys, we have mapped the fine-scale spatiotemporal dynamics of PfEIR and PfPR in a region of Malawi with moderately intense, seasonally variable malaria parasite transmission. We found evident spatial heterogeneity in both PfEIR and PfPR, with areas of higher PfEIR and PfPR expanding and contracting over time. We also found that hotspots of PfEIR and hotspots of PfPR overlapped at times, but the amount of overlap varied over time. Finally, we showed that month-to-month variations in PfEIR over the study period are strongly associated with changes in PfPR. These findings highlight the dynamic nature of malaria parasite transmission and underscore the value of monitoring both PfEIR and PfPR at fine spatial and temporal resolutions.

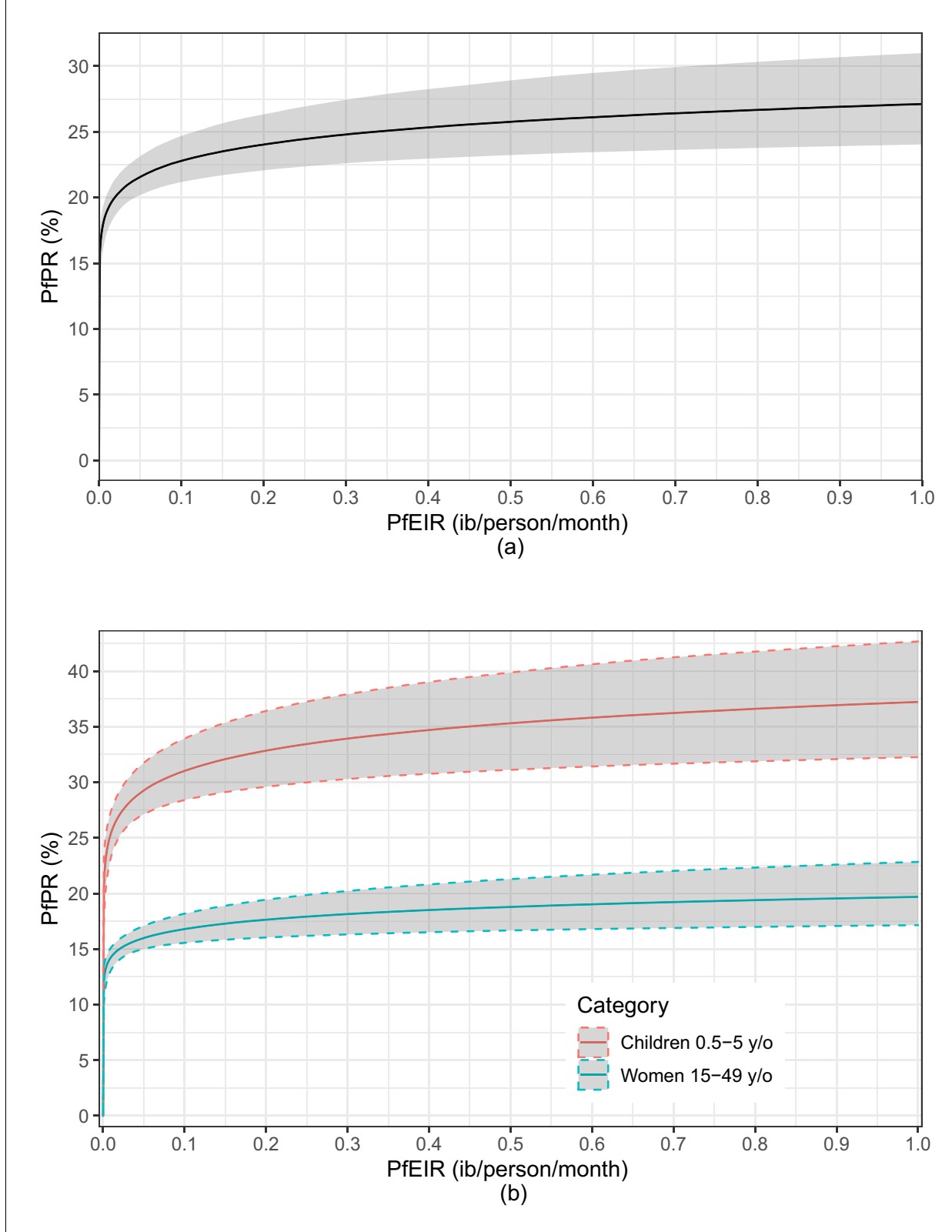

**Figure 8.** A plot of the estimated logit-linear relationship between PfPR and PfEIR. The solid lines are the estimated relationships and the shaded areas are the associated 95% confidence region for children and women combined (a) and for children and women separately (b). The shaded regions represent the corresponding 95% confidence region. The confidence regions were obtain from 10,000 predictive samples where each sample was obtained by plugging in one of the bootstrap samples parameter estimate into the logit-linear model.

In this study, we used model-based geostatistics (*Diggle and Giorgi, 2019*) to analyse repeated cross-sectional data with a unique sampling framework (*Roca-Feltrer et al., 2012*; *Kabaghe et al., 2017*) and estimate the fine-scale spatial patterns of PfEIR and PfPR across our study site at monthly intervals. This approach was essential for identifying hotspots of PfEIR and PfPR through the use of exceedance probabilities, because it allowed us to acknowledge the effects of unmeasured malaria risk factors on each metric through the inclusion of spatially structured random variations. Consequently, we were able to directly compare the spatial patterns of PfEIR and PfPR over an extended period of time in a single geographical region. As expected, there were hotspots identified by each of the two metrics of malaria transmission risk, which expanded and retracted over time. However, the hotspots of PfEIR and PfPR only partially overlapped, with the most substantial amount of overlap observed during the high transmission seasons. Within-village and between-village spatial heterogeneities of malaria parasite transmission are well documented across many sites (*Greenwood, 1989*; *Thompson et al., 1997*; *Amek et al., 2012*; *Mwandagalirwa et al., 2017*), but few previous studies have compared the spatial distributions of different transmission metrics in the same site (but see *Stresman et al., 2017*). Given that all available metrics of malaria parasite transmission have their own strengths and weaknesses (*Tusting et al., 2014*), our findings suggest that monitoring multiple transmission metrics, each aligned with widely separated steps of the parasite transmission cycle, provides a more complete understanding of the underlying spatial heterogeneity in malaria parasite transmission (*Cohen et al., 2017*). Furthermore, monitoring multiple metrics may provide an opportunity to optimise the impact of control activities by targeting different control activities to different locations based on differences in the metrics (*Cohen et al., 2017*). For example, areas with higher transmission risk according to an entomological metric (e.g. PfEIR) than a measure of the potential transmission reservoir (e.g. PfPR) may indicate a need for increased vector control, whereas areas with lower PfEIR and higher PfPR may indicate a need for increased treatment of malaria parasite infections.

Our geostatistical modelling approach also provided a principled framework for estimating PfEIR in our study, allowing us to robustly investigate the PfEIR-PfPR relationship despite the low mosquito densities observed in this region. Previous studies *Beier et al., 1999*; *Smith et al., 2005* have estimated the relationship between PfEIR and PfPR using paired estimates of these metrics from several sites throughout Africa, characterised by a wide range of transmission intensities (PfEIR <1 to >500 ib/person/year). These studies demonstrated that small changes in PfEIR are associated with large changes in PfPR when PfEIR is low, for example below about 15 ib/person/year (*Beier et al., 1999*). However, estimating PfEIR in settings with low mosquito density is challenging because the accuracy and precision of PfEIR depend on the accuracy and precision of the human biting rates and sprorozite rates used to calculate PfEIR (*Tusting et al., 2014*). We overcame these challenges by using model-based geostatistics to improve the precision of our PfEIR estimates and, just as importantly, used bootstrap procedures to propagate the uncertainty from each modelling step to the next. Altogether, these methods allowed us to investigate the PfEIR-PfPR relationship with a focus on much lower ranges of PfEIR than previous studies.

An additional advantage of using geostatistical models in this study was the prediction of entomological data at unsampled geographical locations. For a number of logistical reasons (e.g. mosquito sampling was conducted over 2 consecutive nights at each sampled location), we sampled for mosquitoes at roughly 75% of the locations visited for parasitaemia sampling in each round. For the geographical locations where empirical parasitaemia data were available but entomological data were not, our geostatistical model predictions of PfEIR were combined with the empirical PfPR data. This rigorous statistical solution enriched the data used in our assessment of the PfEIR-PfPR relationship.

Prior to our study, the most recent assessment of PfEIR in this district of Malawi was from 2002, with an estimated annual PfEIR of 183 ib/person/year (*Mzilahowa et al., 2012*). The drastic reduction in annual PfEIR since then to an estimated 2.60 ib/person/year in our study is likely due, at least in part, to an increase in the use of ITNs and ACTs as observed elsewhere (*Bayoh et al., 2010*; *Mwangangi et al., 2013*). Nationwide, use of ITNs by children under 5 years old in Malawi has increased from nil in 2000 and 14.8% in 2004 (*Mathanga et al., 2012*) to 67.8% in 2014 (*Malawi National Malaria Control Programme and ICF International, 2014*). Treatment for malaria in Malawi switched from sulfadoxine–pyrimethamine to ACT with artemether–lumefantrine in 2007 (*Mathanga et al., 2012*), and by 2014, 39.3% of children under five reporting a fever had taken ACT

(*Malawi National Malaria Control Programme and ICF International, 2014*). In addition to these long-term, nationwide trends, changes in malaria intervention coverage over time also likely impacted malaria parasite transmission in the more specific context of our study. Mass distributions of ITNs took place across Malawi in 2012 (*World Health Organization, 2013*), that is about 3 years prior to our study, and again in April 2016, that is, 1 year into our 38-month study. Additionally, randomly selected villages in our study site implemented community-led larval source management, house improvement, or both as part of a randomised trial between May 2016 and May 2018 (*McCann et al., 2017b*; *van den Berg et al., 2018*). Although no differences in PfEIR or PfPR were observed between the trial arms (*McCann et al., 2021*), the entire study site, including the trial's control arm, was included in the 2016 mass ITN distribution, as well as other National Malaria Control Programme interventions and a community engagement programme as part of the Majete Malaria Project (*McCann et al., 2017b*; *van den Berg et al., 2018*). Therefore, the changes in PfEIR and PfPR over time observed in this study reflect the combined effects of seasonal weather patterns, year-to-year climatic variation and all malaria control activities.

Within this context of observing month-to-month changes in both PfEIR and PfPR in a single geographical region, we have demonstrated that fluctuations in PfEIR over short periods are associated with predictable changes in PfPR in the same region. We found that a logit-linear model explained the PfEIR-PfPR relationship better than any of the other five model classes examined, and our data better supported a one-month delayed effect of PfEIR on PfPR than no delayed effect or a 2-month delayed effect. The one-month delayed effect is likely due to the incubation period of the parasite (*Ruan et al., 2008*) and the duration of infections (*Felger et al., 2012*). As shown in previous studies (*Beier et al., 1999*; *Smith et al., 2005*), we observed that small changes in PfEIR led to relatively large changes in PfPR at lower ranges of PfEIR, while PfPR saturated rather than changing at a constant rate at higher ranges of PfEIR. These previous studies were based on estimates of PfEIR and PfPR from 31 locations (*Beier et al., 1999*) and more than 90 locations (*Smith et al., 2005*) representing a wide range of PfEIR and PfPR in Africa. By assuming that the PfEIR-PfPR relationship is constant across space on a continental scale, results from these previous studies suggested that variation between PfEIR and PfPR across geographical location is representative of variation between PfEIR and PfPR over time. In our study, we explicitly confirmed this association between PfEIR and PfPR within a single location. Whilst we also assumed that the PfEIR-PfPR relationship does not change dynamically across space and time, we believe this to be a realistic assumption for the restricted geographical setting of our study.

The saturation in PfPR with increasing PfEIR may be explained by several factors, which are not mutually exclusive. One set of factors relates to people being heterogeneously exposed to vectors (*Guelbéogo et al., 2018*) because of differences in attractiveness (*Knols et al., 1995*; *Qiu et al., 2006*), behaviour (*Sherrard-Smith et al., 2019*; *Finda et al., 2019*), access to ITNs (*Bhatt et al., 2015b*), housing design (*Tusting et al., 2015*; *Tusting et al., 2017*), or the spatial distribution of vector habitat (*McCann et al., 2017a*), so that as PfEIR increases, it is more likely that infectious vectors are biting already infected individuals (*Smith et al., 2007b*; *Smith et al., 2010*). The second set of factors relates to inter-individual variation in acquired immunity, which in some individuals may prevent vector-inoculated sporozoites from progressing to blood-stage infection (*John et al., 2005*; *Offeddu et al., 2017*), keep blood-stage infections at densities lower than the level of detection (*Doolan et al., 2009*) (about 50–200 parasites/µl for RDTs as used in our study), or increase the rate at which blood-stage infections are cleared (*Hviid et al., 2015*).

Regardless of the underlying factors driving the PfEIR-PfPR relationship, our results have practical implications for the selection and interpretation of malaria parasite transmission metrics. In settings with higher ranges of PfEIR, moderate changes in PfEIR will not be associated with appreciable changes in PfPR (*Beier et al., 1999*; *Smith et al., 2005*; *Churcher et al., 2015*). Framed in terms of a public health goal to decrease PfPR in these settings with high baseline PfEIR, relatively large reductions in PfEIR would be required to achieve appreciable reductions in PfPR. In terms of selecting an appropriate metric for monitoring changes in transmission in these same high-PfEIR settings, PfPR may only be suitable for measuring very large changes in transmission.

Conversely, in settings with a lower range of PfEIR, our results show that PfPR is sensitive to smaller, short-term changes in malaria parasite transmission. This finding highlights the importance of sustaining vector control efforts in settings with relatively low PfEIR, because a small increase in the rate of infectious bites (PfEIR) could result in a rapid increase in the proportion of people

infected (PfPR). This sensitivity of PfPR to short-term changes in parasite transmission, when PfEIR is low, also confirms that PfPR can be used for monitoring changes in the intensity of parasite transmission linked to either environmental conditions or the effects of malaria interventions. However, this sensitivity of PfPR in these settings suggests that annual cross-sectional surveys aiming for a transmission peak are more likely to miss the actual peak than in settings with higher parasite transmission intensity, as shown previously by others (*Kang et al., 2018*). National malaria control programs and others planning malaria indicator surveys to measure year-to-year variation in PfPR should therefore consider approaches to identify and account for any potential bias in PfPR estimates from a single time point, for example incorporating continuous or 'rolling' surveys at sentinel sites (*Roca-Feltrer et al., 2012*; *Kabaghe et al., 2017*), monitoring 'easy-access groups' (*Sesay et al., 2017*), or modelling sub-annual trends based on health facility data (*Sturrock et al., 2014*; *Awine et al., 2018*). These considerations likely apply to settings where increasing coverage of ITNs (*Bhatt et al., 2015a*) and ACTs (*Bennett et al., 2017*) has reduced PfPR from ≥50% (i.e. holo- and hyperendemic [*Hay et al., 2008*]) to between 10–50% (i.e. mesoendemic), which have become increasingly common over the last 20 years (*Weiss et al., 2019*).

The monthly PfEIR in our study region was 0 ib/person/month in multiple months. This likely indicates that the number of infectious bites received per person during these months was below the level of detection, rather than an actual interruption of transmission during those months, especially in the first 2 years of the study when these periods only lasted 2–3 months. Our finding that a monthly PfEIR near or equal to zero is associated with substantial PfPR is, therefore, unsurprising given that previous studies have had similar findings when comparing annual PfEIR to PfPR (*Kabiru, 1994*; *Mbogo et al., 1995*; *Beier et al., 1999*; *Smith et al., 2005*). On the other hand, we observed an increase in PfPR from about November 2017 to May 2018 while PfEIR remained at zero. It remains unclear whether this rise in PfPR was due to new infectious bites that nevertheless remained below the level of detection or to previously infected individuals with parasite densities that increased to detectable levels (*Drakeley et al., 2018*). Either way, this result shows that a rise in PfPR may be observed even when PfEIR cannot be detected by current methods, and, therefore, both interventions and monitoring need to continue for some time after PfEIR has not been detected. Our results also highlight the importance of monitoring additional metrics of parasite transmission (in addition to PfEIR) when the annual PfEIR is <10 ib/person/year, especially when expecting a reduction in transmission as in the case of testing malaria interventions. Nonetheless, when PfEIR is above the level of detection, it provides information about the vector species involved in transmission, which is critical because different mosquito species may respond differently to vector control interventions (*Ferguson et al., 2010*; *Wilson et al., 2020*).

We observed a consistently higher PfPR in children 0.5–5 y/o than in women 15–49 y/o throughout the study region and study period, as expected. The extent of difference in PfPR between children and adults for a given region generally increases with parasite transmission intensity. However, even in mesoendemic settings (PfPR between 10–50%), it is common for PfPR in children to be appreciably higher than in adults (*Smith et al., 2007a*). This pattern is due to increasing acquired immunity with increased exposure to malaria parasites over time (*Baird, 1995*), which may decrease transmission efficiency and time to clear a *P. falciparum* infection in adults compared to children (see Appendix 1 – Procedure for building the HBR, PfSR, and PfPR models, *Appendix 1—table 6* and *Smith et al., 2005*). Moreover, the higher PfPR in children than adults, even at the lowest levels of transmission, suggests that children may play a more significant role in transmission, consistent with other studies (*Walldorf et al., 2015*; *Ouédraogo et al., 2016*).

Although the functional form of our best fitting PfEIR-PfPR model matches that of previous studies (*Beier et al., 1999*; *Smith et al., 2005*), the estimated values of PfPR as a function of PfEIR show non-negligible differences. For example, based on our best model, a 0.15 ib/person/month decrease in PfEIR from 0.2 to 0.05 ib/person/month leads to a decrease in PfPR from 24.10% to 21.66% on average, a reduction of about 10.13%. In *Beier et al., 1999*, instead, the same decrease in PfEIR corresponds to a decrease in PfPR from 33.88% to 19.31%, a reduction of about 43.00%, whilst in the case of the best model of *Smith et al., 2005*, that yields a decrease in PfPR from 33.46% to 16.51%, a reduction of about 50.66%. One possible reason for these differences is that our study focuses on a geographically limited area where lower values in PfEIR are observed. Secondly, both previous studies excluded data from sites reporting mosquito control activities whilst our study site included multiple interventions. Finally, another important difference with our study is that, our focus was on

a relatively small sub-national area of Malawi, whereas *Beier et al., 1999* and *Smith et al., 2005* used data from across Africa and implicitly assumed that the properties of the PfEIR-PfPR relationship do not vary over such continental scale.

In this study, the empirical PfPR was used as the response variables of six different statistical models, while the modelled PfEIR from a geostatistical model was used as the independent variable in each of the six models. As shown in this paper, this approach has two main advantages: (1) it allowed us to develop a bootstrap procedure for propagating the uncertainty arising from the estimates of PfEIR into the PfPR-PfEIR relationship; (2) it allowed us to avoid the generation of spurious correlations in the estimation of the PfPR-PfEIR relationship. The risk of spurious correlation may in fact occur when using approaches that are based on the estimates of both PfPR and PfEIR, which are obtained from statistical models informed by the same set of covariates.

One limitation of this study was that the six PfEIR-PfPR models do not allow for overdispersion in the estimation of the PfEIR-PfPR relationship. However, the use of standard Binomial likelihoods still delivers unbiased estimates of the functional relationship between PfEIR and PfPR, even in the presence of overdispersion (*Godambe and Kale, 1991*). Furthermore, given that the uncertainty around the PfPR-PfEIR relationship is mainly driven by the predictive distribution of PfEIR, which we account for through our bootstrap procedure, we do not expect overdispersion to have non-negigible influence on the parameter estimates. Finally, the development of models that allow for overdispersion may be achieved in several different ways, for example, by modelling the parameters that modulate the PfEIR-PfPR relationship as stochastic processes. However, these approaches would require a larger amount of data than those available in this study and should be the subject of future research.

A second limitation was the use of RDTs to estimate PfPR. RDTs can show false positives after anti-malarial drug treatment due to persistence of the antigens detected by RDTs (*Dalrymple et al., 2018*). Also, the limit of detection (usually 50–200 parasites/µl) is higher than that of expert microscopy or PCR (*Chiodini, 2014*). In modelling the relationship between PfEIR and PfPR, we did not account for the sensitivity and specificity of the RDT used to detect *P falciparum* infection. If the sensitivity α and specificity β were known, we could account for them by setting $PfPR(x, t)$ as used in our analysis to $\alpha(PfPR(x,t)\beta - 1)/(\alpha + \beta - 1)$. Thus, strictly, what we have called PfPR should be interpreted as the probability of testing positive for *P. falciparum* using RDT. However, the use of RDTs as a diagnostic test for the detection of malaria infection provides PfPR estimates that are comparable to national malaria indicator surveys.

An additional limitation of our study was the unidirectional relationship implicitly assumed in our models of PfEIR-PfPR. In reality, PfPR and PfEIR are causally linked by the malaria parasite transmission cycle, which alternates between the human host and the mosquito vector. A higher rate of infectious bites received per person (i.e. PfEIR) increases the probability of the person becoming infected when bitten. Therefore, any factor that reduces mosquito populations, biting rates or human-to-mosquito parasite transmission (e.g. effective vector control interventions) will reduce PfEIR and consequently translate to reductions in PfPR. Similarly, a higher rate of parasite infection in people (i.e. PfPR) increases the probability of a mosquito becoming infected after any given blood meal. Therefore, factors that directly reduce PfPR (e.g. treatment of infections with effective drugs) will consequently reduce PfEIR. The impact of interventions may therefore affect both PfEIR and PfPR in such a way that a cyclic relationship may better describe the association between these metrics. Future modelling efforts may thus be improved by taking into account the cyclic aspect of the PfEIR-PfPR relationship.

## Conclusion

Measuring PfEIR and PfPR using the rolling MIS sampling framework and a geostatistical modelling approach allowed us to assess the fine-scale spatial and temporal distributions of malaria parasite transmission over 38 months in a mesoendemic setting. The relationship between PfEIR and PfPR estimated here shows that at low levels of PfEIR, changes in PfEIR are associated with rapid changes in PfPR, while at higher levels of PfEIR, changes in PfEIR are not associated with appreciable changes in PfPR. Comparing hotspots of PfEIR and PfPR revealed that each metric could identify potential transmission hotspots that the other fails to capture. Our results emphasise that PfEIR and PfPR are essential, complementary metrics for monitoring short term changes in *P. falciparum* transmission intensity in mesoendemic settings, which have become increasingly common as many regions reduce transmission and shift from the highest malaria endemicity levels. Our study emphasises the need to

couple vector control with identifying and treating infected individuals to drive malaria to elimination levels and to monitor both entomological and parasitaemia indices in malaria surveillance.

## Acknowledgements

This study was generously supported by Dioraphte Foundation, The Netherlands. RSM received additional support from the NIH (T32AI007524 and K01TW011770). The content is solely the responsibility of the authors and does not necessarily represent the official views of the funders. We thank African Parks and The Hunger Project for their significant and practical contributions in facilitating the study. We are grateful to the entire Majete Malaria Project team for their tireless efforts in carrying out the study. The population of the study area is thanked for their cooperation with the study. We also thank: Alexandra Hiscox for advice on mosquito sampling and study design; Jeroen Spitzen for logistical assistance; and Martin Donnelly, Karl Seydel, and their respective laboratory teams for assistance in molecular identification of malaria parasites and anopheline mosquitoes.

## Additional information

### Funding

| Funder | Grant reference number | Author |
| --- | --- | --- |
| Stichting Dioraphte | 13050800 | Willem Takken |
| National Institutes of Health | T32AI007524 | Robert S McCann |
| National Institutes of Health | K01TW011770 | Robert S McCann |

The funders had no role in study design, data collection and interpretation, or the decision to submit the work for publication.

### Author contributions

Benjamin Amoah, Conceptualization, Software, Formal analysis, Validation, Investigation, Visualization, Methodology, Writing - original draft, Writing - review and editing; Robert S McCann, Conceptualization, Formal analysis, Investigation, Methodology, Writing - original draft, Writing - review and editing; Alinune N Kabaghe, Formal analysis, Methodology, Writing - original draft, Writing - review and editing; Monicah Mburu, Formal analysis, Writing - original draft, Writing - review and editing; Michael G Chipeta, Steven Gowelo, Tinashe Tizifa, Themba Mzilahowa, Michele van Vugt, Investigation, Writing - original draft, Writing - review and editing; Paula Moraga, Formal analysis, Investigation, Writing - original draft, Writing - review and editing; Henk van den Berg, Willem Takken, Kamija S Phiri, Conceptualization, Funding acquisition, Investigation, Writing - original draft, Writing - review and editing; Peter J Diggle, Conceptualization, Formal analysis, Supervision, Validation, Investigation, Methodology, Writing - original draft, Writing - review and editing; Dianne J Terlouw, Conceptualization, Supervision, Funding acquisition, Investigation, Methodology, Writing - original draft, Writing - review and editing; Emanuele Giorgi, Formal analysis, Supervision, Methodology, Writing - original draft, Writing - review and editing

### Author ORCIDs

Benjamin Amoah ⬡ https://orcid.org/0000-0002-9650-2704
Robert S McCann ⬡ https://orcid.org/0000-0002-2195-6461
Michael G Chipeta ⬡ https://orcid.org/0000-0001-5882-9936
Henk van den Berg ⬡ https://orcid.org/0000-0002-9983-638X
Emanuele Giorgi ⬡ https://orcid.org/0000-0003-0640-181X

### Ethics

Human subjects: Written informed consent was sought from individual participants (and their parents or guardians when appropriate) during household surveys. The College of Medicine Research and Ethics Committee in Malawi approved this study (proposal number P.05/15/1731).

Decision letter and Author response
Decision letter https://doi.org/10.7554/eLife.65682.sa1
Author response https://doi.org/10.7554/eLife.65682.sa2

# Additional files

## Supplementary files

- Transparent reporting form

## Data availability

The data are available at Lancaster University Library online (https://doi.org/10.17635/lancaster/researchdata/479). Geolocations of individuals have been removed to anonymise the data. However, geolocation can be assessed by sending proposals to Professor Michèle van Vugt (m.vanvugt@amsterdamumc.nl). Data requestors will need to apply for ethical clearance and sign a data access agreement.

The following dataset was generated:

| Author(s) | Year | Dataset title | Dataset URL | Database and Identifier |
|---|---|---|---|---|
| Giorgi E | 2021 | Entomology and parasitaemia data | https://doi.org/10.17635/lancaster/researchdata/479 | Lancaster University Research Data, 10.17635/lancaster/researchdata/479 |

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

## Appendix 1

### Procedure for building the HBR, PfSR and PfPR models

Let $Avg(\text{Temp}(x_i, t_i), s_1, s_2)$ and $Avg(\text{RH}(x_i, t_i), s_1, s_2)$ respectively denote the average temperature and relative humidity taken over $s_1$ to $s_2$ days prior to the data collection. Procedure for building the HBR, PfSR and PfPR models *Appendix 1—table 1* shows the $s_1$ and $s_2$ values over which average temperature and relative humidity were computed. A set of these variables were selected as the best predictors each of the outcome variables based on the procedure in the next section.

We selected the best combination of fixed and random effects that best explain HBR, PfSR and PfPR using the following procedure.

1. We first built a generalized linear model in which temperature and RH are considered together with time trends and sine and cosine functions for seasonality. For $Avg(\text{Temp}(x_i, t_i), s_1, s_2)$, $Avg(\text{RH}(x_i, t_i), s_1, s_2)$, the choice of $s_1$ and $s_2$, as illustrated by Procedure for building the HBR, PfSR and PfPR models *Appendix 1—table 1*, was based on the deviance profile of the variable involved, that is, either temperature or RH. Piecewise-linear transformations of temperature and RH were considered based on visual inspection and epidemiological knowledge.
2. Potential confounding between seasonal sinusoids, temperature and RH were checked. Covariates that did not improve the model fit as judged by the AIC were excluded. Sin-cosine terms were always considered together as if they were one covariate.
3. With the current model as a basic model we include other available explanatory variables based on forward selection, and check for interactions.
4. When no more explanatory variables significantly improve the model fit, we fit a generalized linear mixed model with a random effect for each unique space-time location.
5. We then check for the presence of residual spatial, temporal, and spatio-temporal correlations using the algorithm described in *Giorgi et al., 2018*, and then include the random effect terms that improve the model fit.

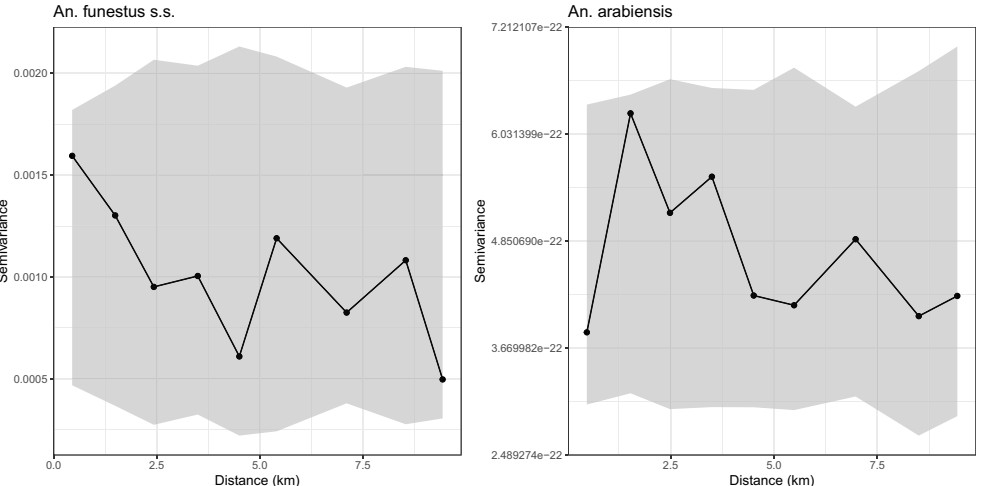

**Appendix 1—figure 1.** Empirical variograms (Solid lines) and 95% confidence regions (grey areas) developed by randomly pairing locations, that is, assuming that there is no residual spatial correlation in the sporoziote rate data of *An funestus s.s.* (left panel) and *An arabiensis* (right panel). The variograms lie entirely within the respective 95% confidence regions, indicating that there is no residual spatial correlation.

### The selected fixed effects for the HBR, PfSR and PfPR models

We specify the set of fixed effects we selected to be in the final model for the *A. arabiensis* HBR, *A. funestus s.s.* HBR, PfSR, and the PfPR models. Detailed description of the terms involved in the fixed

effects and the estimates of all the parameters of each model are given in S1 *Appendix 1—table 2–5*.

- *A. arabiensis* human biting rate

$$
\begin{aligned}
d(x_i,t_i)^\top\beta + f(t_i;\alpha) = \ & \beta_1\mathbf{1}(x_i \in \mathcal{A}) + \beta_2\mathbf{1}(x_i \in \mathcal{B}) + \beta_3\mathbf{1}(x_i \in \mathcal{C}) + \beta_4\mathbf{1}(\text{Indoor}) + \\
& \beta_5\text{DSR}(x_i) + \beta_6 Avg(\text{RH}(x_i,t_i),14,35) + \\
& \beta_7\min\{Avg(\text{Temp}(x_i,t_i),7,14),22.9\} \\
& +\beta_8\max\{Avg(\text{Temp}(x_i,t_i),7,14) - 22.9,0\} + \\
& \alpha_1\sin(2\pi t_i/12)/t + \alpha_2\cos(2\pi t_i/12)/t
\end{aligned}
$$

- *A. funestus s.s.* human biting rate

$$
\begin{aligned}
d(x_i,t_i)^\top\beta + f(t_i;\alpha) = \ & \beta_0 + \text{Elevation}(x_i) + \beta_1\text{DSR}(x_i) + \beta_2\text{NDVI}(x_i) + \\
& \beta_3 Avg(\text{Temp}(x_i,t_i),0,7) + \beta_4 Avg(\text{Temp}(x_i,t_i),7,14) + \\
& +\beta_5 Avg(\text{RH}(x_i,t_i),14,21) \\
& +\alpha_1\sin(2\pi t_i/12) + \alpha_2\cos(2\pi t_i/12) + \\
& +\alpha_3\min\{t_i,12\} + \alpha_4\max\{t_i - 12,0\}
\end{aligned}
$$

- *A. arabiensis* sporozoite rate

$$
\begin{aligned}
d(x_i,t_i)^\top\beta^* + f^*(t_i;\alpha^*) = \ & \beta_0^* + \beta_1^*\text{DLR}(x_i) + \beta_2^*\text{DSR}(x_i) + \beta_3^*\text{Elevation}(x_i) + \\
& \beta_4^*\text{EVI}(x_i) + \alpha_1^*\sin(2\pi t_i/12) + \alpha_2^*\cos(2\pi t_i/12) + \\
& \alpha_3^*\min\{t_i,12\} + \alpha_4^*\max\{t_i - 12,0\}
\end{aligned}
$$

- *A. funestus s.s.* sporozoite rate

$$
\begin{aligned}
d(x_i,t_i)^\top\beta^* + f^*(t_i;\alpha^*) = \ & \beta_0^* + \alpha_1^*\sin(2\pi t_i/12) + \alpha_2^*\cos(2\pi t_i/12) + \\
& \alpha_3^*\min\{t_i,12\} + \alpha_4^*\max\{t_i - 12,0\}
\end{aligned}
$$

- *P. faciparum* prevalence

$$
\begin{aligned}
d(x_i,t_i)^\top\varphi + g(t_i;\varrho) = \ & \varphi_1\mathbf{1}(x_i \in \mathcal{A}) + \varphi_2\mathbf{1}(x_i \in \mathcal{B}) + \varphi_3\mathbf{1}(x_i \in \mathcal{C}) + \\
& \varphi_4\text{Elevation}(x_i) + \varphi_5\text{DLR}(x_i) + \\
& \varphi_6 Avg(\text{Temp}(x_i,t_i),14,42) + \varphi_7\text{NDVI}(x_i) + \\
& \varphi_8\text{Wealth}(x_i) + \varrho_1\min\{t_i,21\} + \varrho_2\max\{t_i - 21,0\} + \\
& +\varrho_3\cos(2\pi t_i/12) + \varrho_4\sin(2\pi t_i/12)
\end{aligned}
$$

**Appendix 1—table 1.** Range of days prior to data collections over which temperature and relative humidity were averaged.

| To $(s_2)$ | 0 | 3 | 5 | 7 | 14 | 21 | 28 | 35 | 42 |
|---|---|---|---|---|---|---|---|---|---|
| **From** $(s_1)$ | | | | | | | | | |
| 0 | | ✓* | ✓ | ✓ | ✓ | ✓ | ✓ | ✓ | ✓ |
| 3 | | | ✓ | ✓ | ✓ | ✓ | ✓ | ✓ | ✓ |
| 5 | | | | ✓ | ✓ | ✓ | ✓ | ✓ | ✓ |
| 7 | | | | | ✓ | ✓ | ✓ | ✓ | ✓ |
| 14 | | | | | | ✓ | ✓ | ✓ | ✓ |
| 21 | | | | | | | ✓ | ✓ | ✓ |
| 28 | | | | | | | | ✓ | ✓ |
| 35 | | | | | | | | | ✓ |

*The check marks indicate the days from/to which temperature and relative humidity were averaged.

**Appendix 1—table 2.** Regression table for the *A. arabiensis* human biting rate model.

| Variable | Description | Parameter | Point estimate |
|---|---|---|---|
| *Covariates* | | | |
| $\mathbf{1}(x_i \in \mathcal{A})$ | A binary indicator taking the value 1 if location $x_i$ | $\beta_1$ | −13.525 |
| | belongs to Focal Area A and 0 otherwise. | | (−16.217,–10.833)* |
| $\mathbf{1}(x_i \in \mathcal{B})$ | A binary indicator taking the value 1 if location $x_i$ | $\beta_2$ | −9.995 |
| | belongs to Focal Area B and 0 otherwise. | | (−12.656,–7.333) |
| $\mathbf{1}(x_i \in \mathcal{C})$ | A binary indicator taking the value 1 if location $x_i$ | $\beta_3$ | −10.848 |
| | belongs to Focal Area C and 0 otherwise. | | (−13.514,–8.182) |
| $\mathbf{1}(\text{Indoor})$ | A binary indicator taking the value 1 if the mosquito | $\beta_4$ | 0.456 |
| | trap was set indoors and 0 otherwise. | | (0.264, 0.647) |
| $\text{DSR}(x_i)$ | Distance from location $x_i$ to the closest small river | $\beta_5$ | $0.631 \times 10^{-3}$ |
| | | | $( 0.143, 1.120 ) \times 10^{-3}$ |
| $Avg(\text{RH}(x_i, t_i), 14, 35)$ | Average relative humidity 14 to 35 days prior to the | $\beta_6$ | 0.056 |
| | data collection. | | (0.038, 0.073) |
| $\min\{Avg(\text{Temp}(x_i, t_i), 7, 14), 22.9\}$ | The effect of temperature when temperature is | $\beta_7$ | 0.180 |
| | below 22.9°C. | | (0.072, 0.289) |
| $\max\{Avg(\text{Temp}(x_i, t_i), 7, 14) - 22.9, 0\}$ | The effect of temperature when temperature is | $\beta_8$ | −0.132 |
| | 22.9°C or higher. | | (−0.22,–0.044) |
| *Seasonality and Trends* | | | |
| $\sin(2\pi t_i/12)/t$ | | $\alpha_1$ | −0.291 |
| | | | (−0.907, 0.325) |
| $\cos(2\pi t_i/12)/t$ | | $\alpha_2$ | 1.092 |
| | | | (−0.759, 2.943) |
| *Spatial Correlation* | | | |
| Signal variance | | $\sigma^2$ | 4.114 |
| | | | (3.262, 5.189) |
| Scale (km) | | $\phi$ | 0.649 |
| | | | (0.492, 0.856) |
| Nugget variance | | $\tau^2$ | 0.162 |
| | | | (0.124, 0.21) |

Dependent Variable: log of A. funestus mosquito density.

*95% confidence intervals are in brackets.

**Appendix 1—table 3.** Regression table for the *A. funestus* human biting rate model.

| Variable | Description | Parameter | Point estimate |
|---|---|---|---|
| *Covariates* | | | |

*Continued on next page*

*Appendix 1—table 3 continued*

| Variable | Description | Parameter | Point estimate |
|---|---|---|---|
| Intercept | | $\beta_0$ | 2.523 |
| | | | (−3.209, 8.256)* |
| Elevation($x_i$) | Elevation of the location $x_i$. | $\beta_1$ | $-5.583 \times 10^{-3}$ |
| | | | $(-7.896, -3.271) \times 10^{-3}$ |
| DSR($x_i$) | Distance from location $x_i$ to the nearest small river. | $\beta_2$ | $2.993 \times 10^{-3}$ |
| | | | $(2.329, 3.658) \times 10^{-3}$ |
| NDVI($x_i$) | Normalized difference vegetation index at location $x_i$. | $\beta_3$ | 1.392 |
| | | | (−1.251, 4.035) |
| $Avg(\text{Temp}(x_i, t_i), 0, 7)$ | Average temperature one week prior to data collection. | $\beta_4$ | −0.154 |
| | | | (−0.279, −0.028) |
| $Avg(\text{Temp}(x_i, t_i), 7, 14)$ | Average temperature 7 to 14 days prior to data collection. | $\beta_5$ | −0.116 |
| | | | (−0.295, 0.064) |
| $Avg(\text{RH}(x_i, t_i), 14, 21)$ | Average relative humidity 14 to 21 days prior to data collection. | $\beta_6$ | −0.043 |
| | | | (−0.078, −0.008) |
| *Seasonality and Trends* | | | |
| $\sin(2\pi t_i/12)$ | | $\alpha_1$ | −0.291 |
| | | | (−0.907, 0.325) |
| $\cos(2\pi t_i/12)$ | | $\alpha_2$ | 1.092 |
| | | | (−0.759, 2.943) |
| $\min\{t_i, 12\}$ | | $\alpha_3$ | −0.291 |
| | | | (−0.907, 0.325) |
| $\max\{t_i - 12, 0\}$ | | $\alpha_4$ | 1.092 |
| | | | (−0.759, 2.943) |
| *Spatial Correlation* | | | |
| Signal variance | | $\sigma^2$ | 4.456 |
| | | | (3.379, 5.876) |
| Scale (km) | | $\phi$ | 0.906 |
| | | | (0.66, 1.245) |
| Nugget variance | | $\tau^2$ | 0.142 |
| | | | (0.105, 0.191) |

Dependent Variable: log of A. funestus mosquito density

*95% confidence intervals are in brackets.

**Appendix 1—table 4.** Regression table from fitting the *P. falciparum* sporozoite rate models.

| Variable | Description | Parameter | A. funestus s.s. | A. arabiensis |
|---|---|---|---|---|
| *Covariates* | | | | |
| Intercept | | $\beta_0^*$ | 0.139 | −3.392 |
| | | | (−7.793, 8.071)* | (−4.772, −2.125) |
| DLR($x_i$) | Distance from location $x_i$ to the nearest small river. | $\beta_1^*$ | $-1.945 \times 10^{-3}$ | — |
| | | | $(-3.345, -0.545) \times 10^{-3}$ | |
| DSR($x_i$) | Distance from location $x_i$ to the nearest large river. | $\beta_2^*$ | $-4.309 \times 10^{-3}$ | — |
| | | | (−7.499, −1.119) | |

*Continued on next page*

*Appendix 1—table 4 continued*

| Variable | Description | Parameter | A. funestus s.s. | A. arabiensis |
|---|---|---|---|---|
| Elevation($x_i$) | Elevation of location $x_i$. | $\beta_3^*$ | $7.786 \times 10^{-3}$ | — |
| | | | $(5.819, 9.752) \times 10^{-3}$ | |
| EVI($x_i$) | Enhanced vegetation index of location $x_i$. | $\beta_4^*$ | $-36.648$ | — |
| | | | $(-65.090, -8.206)$ | |
| *Seasonality* | | | | |
| *and Trends* | | | | |
| $\sin(2\pi t_i/12)$ | | $\alpha_1^*$ | $-0.378$ | $-0.253$ |
| | | | $(-0.565, -0.19)$ | $(-0.882, 0.375)$ |
| $\cos(2\pi t_i/12)$ | | $\alpha_2^*$ | $-0.722$ | $-0.867$ |
| | | | $(-0.954, -0.489)$ | $(-1.864, 0.13)$ |
| $\min\{t_i, 12\}$ | | $\alpha_3^*$ | $-0.056$ | $0.027$ |
| | | | $(-0.072, -0.041)$ | $(-0.086, 0.140)$ |
| $\max\{t_i - 12, 0\}$ | | $\alpha_4^*$ | $0.061$ | $-0.089$ |
| | | | $(0.039, 0.084)$ | $(-0.305, 0.127)$ |

Dependent Variables: logits of the probability that a mosquito tests positive for sporozoites

*95% confidence intervals are in brackets.

**Appendix 1—table 5.** Regression table for the *P. falciparum* parasite rate model.

| Variable | Description | Parameter | Children under 5 Y/o | Women 15-49 Y/o |
|---|---|---|---|---|
| *Covariates* | | | | |
| $\mathbf{1}(x_i \in \mathcal{A})$ | A binary indicator taking the value 1 if | $\varphi_1$ | $0.685$ | $-0.506$ |
| | $x_i$ belongs to Focal Area A and 0 otherwise. | | $(-1.877, 3.247)$ | $(-3.166, 2.155)$ |
| $\mathbf{1}(x_i \in \mathcal{B})$ | A binary indicator taking the value 1 if | $\varphi_2$ | $2.829$ | $2.568$ |
| | $x_i$ belongs to Focal Area B and 0 otherwise. | | $(0.41, 5.248)$ | $(0.134, 5.002)$ |
| $\mathbf{1}(x_i \in \mathcal{C})$ | A binary indicator taking the value 1 if | $\varphi_3$ | $3.192$ | $2.641$ |
| | $x_i$ belongs to Focal Area C and 0 otherwise. | | $(0.806, 5.577)$ | $(0.224, 5.058)$ |
| Elevation($x_i$) | Elevation of the location $x_i$. | $\varphi_4$ | $5.165 \times 10^{-3}$ | $5.920 \times 10^{-3}$ |
| | | | $(2.322, 8.008) \times 10^{-3}$ | $(3.039, 8.800) \times 10^{-3}$ |
| DLR($x_i$) | Distance from location $x_i$ to the nearest | $\varphi_5$ | $-0.372 \times 10^{-3}$ | $-0.181 \times 10^{-3}$ |
| | large river. | | $(-0.522, -0.222) \times 10^{-3}$ | $(-0.353, -0.009) \times 10^{-3}$ |
| $Avg(\text{Temp}(x_i, t_i), 14, 42)$ | Average temperature 14 to 42 days prior to | $\varphi_6$ | $-0.112$ | $-0.096$ |
| | data collection. | | $(-0.201, -0.023)$ | $(-0.187, -0.005)$ |
| NDVI($x_i$) | Normalized difference vegetation index at | $\varphi_7$ | $-2.424$ | $-5.556$ |
| | location $x_i$. | | $(-4.703, -0.144)$ | $(-7.63, -3.482)$ |
| Wealth($x_i$) | Wealth index of the $i$-th household. | $\varphi_8$ | $-0.212$ | $-0.159$ |

*Continued on next page*

*Appendix 1—table 5 continued*

| Variable | Description | Parameter | Children under 5 Y/o | Women 15-49 Y/o |
|---|---|---|---|---|
| | | | ( −0.283 , −0.141 ) | ( −0.215 , −0.102 ) |
| *Seasonality* | | | | |
| *and Trends* | | | | |
| $\min\{t_i, 21\}$ | | $\varrho_1$ | −0.079 | −0.079 |
| | | | ( −0.098 , −0.06 ) | ( −0.1 , −0.059 ) |
| $\max\{t_i - 21, 0\}$ | | $\varrho_2$ | 0.072 | 0.086 |
| | | | ( 0.042 , 0.102 ) | ( 0.056 , 0.117 ) |
| $\cos(2\pi t_i/12)$ | | $\varrho_3$ | −0.045 | 0.101 |
| | | | ( −0.265 , 0.175 ) | ( −0.123 , 0.324 ) |
| $\sin(2\pi t_i/12)$ | | $\varrho_4$ | 0.209 | 0.175 |
| | | | ( −0.138 , 0.556 ) | ( −0.173 , 0.523 ) |
| *Spatial Correlation* | | | | |
| Signal variance | | $\sigma^2$ | 0.347 | 0.602 |
| | | | ( 0.222 , 0.542 ) | ( 0.416 , 0.872 ) |
| Scale (km) | | $\phi$ | 1.175 | 1.055 |
| | | | ( 0.617 , 2.238 ) | ( 0.631 , 1.765 ) |
| Nugget variance | | $\tau^2$ | 1.546 | 1.368 |
| | | | ( 0.956 , 2.500) | ( 0.932 , 2.007 ) |

S.I. denotes supper infection and D.I/R denotes different infection/recovery rates for children and women. 95% confidence intervals are in brackets. RMSE is the root-mean-square error.

**Appendix 1—table 6.** Parameter estimates from the models for the relationship between PfEIR and PfPR where PfEIR has a month's lag effect on PfPR.
The models' predictive abilities are assessed by the root-mean-square error (RMSE) and bias.

| Model | $p(x,t)$ | $\gamma$ | $\gamma_1$ | $\gamma_2$ | RMSE | Bias |
|---|---|---|---|---|---|---|
| 1. SIS | $\frac{\gamma PfEIR(x,t-1)}{\gamma PfEIR(x,t-1)+1}$ | 7.02 | | | 0.361 | $7.597 \times 10^{-3}$ |
| | | (3.906, 12.284) | | | | |
| 2. SIS with D.I/R | $\sum_{k=1}^{2} \xi_{k,it} \frac{\gamma_k PfEIR(x,t-1)}{\gamma_k PfEIR(x,t-1)+1}$ | | 107.208 | 0.762 | 0.353 | $27.386 \times 10^{-3}$ |
| | | | (0.088, 381.139) | (0.485, 24.344) | | |
| 3. SIS with S.I. | $1 - e^{-\gamma PfEIR(x,t-1)}$ | 1.728 | | | 0.351 | $78.301 \times 10^{-3}$ |
| | | (0.638, 3.087) | | | | |
| 4. SIS with S.I. and D.I/R | $\sum_{k=1}^{2} \xi_{k,it}\left(1 - e^{-\gamma_k PfEIR(x,t-1)}\right)$ | | 22.603 | 0.471 | 0.392 | $99.390 \times 10^{-3}$ |
| | | | (0.128, 67.048) | (0.234, 7.02) | | |
| | | | a | b | | |
| 5. Beier | $a + b\log(PfEIR(x, t-1))$ | | 0.253 | 0.013 | 0.328 | $5.376 \times 10^{-3}$ |
| | | | (0.232, 0.283) | (0.009, 0.021) | | |

*Continued on next page*

*Appendix 1—table 6 continued*

| Model | $p(x, t)$ | $\gamma$ | $\gamma_1$ | $\gamma_2$ | RMSE | Bias |
|---|---|---|---|---|---|---|
| 6. Logit-linear | $\frac{PfEIR(x,t-1)^b}{PfEIR(x,t-1)^b + \exp(-a)}$ | | -0.986 | 0.100 | 0.327 | $4.874 \times 10^{-3}$ |
| | | | (-1.160, -0.804) | (0.062, 0.147) | | |
| Logit-linear for children only | | | -0.523 | 0.119 | | |
| | | | (-0.742, -0.296) | (0.073, 0.174) | | |
| Logit-linear for women only | | | -1.427 | 0.083 | | |
| | | | (-1.575, -1.218) | (0.046, 0.133) | | |

