## [Decision Letter]

**Acceptance summary:**

Using an impressive dataset and a suit of advanced statistical models, the authors illustrated the spatial-temporal patterns of parasite prevalence (PR) and the entomological inoculation rate (EIR) in a rural area in Malawi. This research approach highlights the importance of integrated control malaria strategies based on both entomological and epidemiological surveys.

**Decision letter after peer review:**

Thank you for submitting your article "Identifying *Plasmodium falciparum* transmission patterns through parasite prevalence and entomological inoculation rate" for consideration by *eLife*. Your article has been reviewed by 3 peer reviewers, and the evaluation has been overseen by a Reviewing Editor and Dominique Soldati-Favre as the Senior Editor. The following individuals involved in review of your submission have agreed to reveal their identity: Chih-hao Hsieh (Reviewer #1); Patrick Brown (Reviewer #2).

Essential revisions:

1. The total number of households under survey is M=2432. The total number of survey months = 38. However, if I understand correctly, not every household is surveyed in each month. However, according to the explanation in the Data section, I cannot understand how many households were surveyed for each month. The data structure needs to be illustrated better so that the readers can get better sense out of the Geostatistical Model section. One simple solution is to make animation that shows the sampling sites for each month. The authors can include the animation as Supplementary Information (can be included in the web portal they provided).

2. How to deal with the uncertainty of PfEIR and PfPR in later analyses? Particularly, error propagates through HBRf (x, t)*PfSRf(x, t)l(t) and PfEIR(x,t) = PfEIRf(x,t) + PfEIRa(x,t). Several estimations have this error propagation issue. I am sure that authors can do these properly using bootstrap. Please explain the procedure clearly and earlier in their Geostatistical analysis section. The bootstrapped approach described in the later section is a good solution.

3. Please provide a clear demonstration of the extra value brought by the spatio-temporal model in terms of: (i) space – how are the spatial of the aspects of the association better characterized; and (ii) time – how does the relationship between PfEIR and PfPR vary dynamically over time.

4. On page 6, it would be worth mentioning that frequentist inference was used to estimate model parameters, many readers might associate geostatistical models as Bayesian.

5. The formula on p.16 for calculating PfPR while accounting for sensitivity and specificity seems to be wrong. Shouldn't it be (PfPR + β – 1)/(α + β – 1)? If the formula in the manuscript is correct, please provide a reference.

6. Figure 1: Attribution is required for the map figure (are the raster tiles from Google, OpenStreetMap, or other sources?)

7. Figure 4: is the confidence region calculated with error propagation of estimates of PfSR and PfEIR? Likewise, the uncertainty associated with estimates needs to be carefully addressed in all estimates and clearly depicted in the figures.

8. While the online visualisation is good, the manuscript could include static map figures of the results (PfEIR and PfPR). For example, select certain time points with combination of high/low PfEIR and high/low PfPR. This is useful for archive purposes, and also because only the maps in the manuscript can be said to have been peer-reviewed.

*Reviewer #1 (Recommendations for the authors):*

1. The total number of households under survey is M=2432. The total number of survey months = 38. However, if I understand correctly, not every household is surveyed in each month. However, according to the explanation in the Data section, I cannot understand how many households were surveyed for each month. The data structure needs to be illustrated better so that the readers can get better sense out of the Geostatistical Model section. One simple solution is to make animation that shows the sampling sites for each month. The authors can include the animation as Supplementary Information (can be included in the web portal they provided).

2. How to deal with the uncertainty of PfEIR and PfPR in later analyses? Particularly, error propagates through HBRf (x, t)*PfSRf(x, t)l(t) and PfEIR(x,t) = PfEIRf(x,t) + PfEIRa(x,t). Several estimations have this error propagation issue. I am sure that authors can do these properly using bootstrap. Please explain the procedure clearly and earlier in their Geostatistical analysis section. The bootstrapped approach described in the later section is a good solution.

3. Figure 4: is the confidence region calculated with error propagation of estimates of PfSR and PfEIR? Likewise, the uncertainty associated with estimates needs to be carefully addressed in all estimates and clearly depicted in the figures.

*Reviewer #2 (Recommendations for the authors):*

On page 6, it would be worth mentioning that frequentist inference was used to estimate model parameters, many readers might associate geostatistical models as Bayesian.

The formula on p.16 for calculating PfPR while accounting for sensitivity and specificity seems to be wrong. Shouldn't it be (PfPR + β – 1)/(α + β – 1)? If the formula in the manuscript is correct, please provide a reference.

Figures and Tables:

Figure 1: Attribution is required for the map figure (are the raster tiles from Google, OpenStreetMap, or other sources?)

While the online visualisation is good, the manuscript could include static map figures of the results (PfEIR and PfPR). For example, select certain time points with combination of high/low PfEIR and high/low PfPR. This is useful for archive purposes, and also because only the maps in the manuscript can be said to have been peer-reviewed.

*Reviewer #3 (Recommendations for the authors):*

To strengthen this work, I would want a clear demonstration of the extra value brought by the spatio-temporal model in terms of: (i) space – how are the spatial of the aspects of the association better characterized; and (ii) time – how does the relationship between PfEIR and PfPR vary dynamically over time.

[Editors' note: further revisions were suggested prior to acceptance, as described below.]

Thank you for resubmitting your work entitled "Identifying *Plasmodium falciparum* transmission patterns through parasite prevalence and entomological inoculation rate" for further consideration by *eLife*. Your revised article has been evaluated by Dominique Soldati-Favre (Senior Editor) and a Reviewing Editor.

The manuscript has been improved but there are some remaining issues that need to be addressed, as outlined below:

1. Highlight and clarify in the main text the issue related to potential spurious correlation (statistical artifact) between the modelled PfPR and PfEIR.

2. Address in the main text how the inclusion of an over dispersion term will change the results of modelling the relationship between PfPR and PfEIR in the six models.

3. Introduce into the main text comparison of the estimation PfEIR vs PfPR relationship with previously published models from the literature and explain the observed differences beyond the differences between the modelling approaches.

---

## [Author Response]

Essential revisions:1. The total number of households under survey is M=2432. The total number of survey months = 38. However, if I understand correctly, not every household is surveyed in each month. However, according to the explanation in the Data section, I cannot understand how many households were surveyed for each month. The data structure needs to be illustrated better so that the readers can get better sense out of the Geostatistical Model section. One simple solution is to make animation that shows the sampling sites for each month. The authors can include the animation as Supplementary Information (can be included in the web portal they provided).

We have included two plots as Figure 2 in the Results section (rMIS and mosquito sampling subsection), which indicate how the number of rMIS participants and houses for mosquito samplings were distributed over the course of the study. We have also added a sentence to the Data subsection of Methods as further clarification.

2. How to deal with the uncertainty of PfEIR and PfPR in later analyses? Particularly, error propagates through HBRf (x, t)*PfSRf(x, t)l(t) and PfEIR(x,t) = PfEIRf(x,t) + PfEIRa(x,t). Several estimations have this error propagation issue. I am sure that authors can do these properly using bootstrap. Please explain the procedure clearly and earlier in their Geostatistical analysis section. The bootstrapped approach described in the later section is a good solution.

We used bootstrapping as the reviewer suggests. We have added text to the *Estimating the Plasmodium falciparum entomological inoculation rate* section to provide this further clarification and explanation.

3. Please provide a clear demonstration of the extra value brought by the spatio-temporal model in terms of: (i) space – how are the spatial of the aspects of the association better characterized; and (ii) time – how does the relationship between PfEIR and PfPR vary dynamically over time.

Thanks to the reviewer for raising this point, which we hope will clarify our assumptions about the PfEIR-PfPR relationship and highlight the contributions of this paper. We have made a number of significant changes to the Discussion section to address these points, most notably paragraphs number 2-4 and 6.

4. On page 6, it would be worth mentioning that frequentist inference was used to estimate model parameters, many readers might associate geostatistical models as Bayesian.

We have clarified that parameters were estimated using Monte Carlo Maximum Likelihood and added relevant references.

5. The formula on p.16 for calculating PfPR while accounting for sensitivity and specificity seems to be wrong. Shouldn't it be (PfPR + β – 1)/(α + β – 1)? If the formula in the manuscript is correct, please provide a reference.

The reviewer is correct. This was a mistake on our part, and we have corrected this as suggested.

6. Figure 1: Attribution is required for the map figure (are the raster tiles from Google, OpenStreetMap, or other sources?)

Done.

7. Figure 4: is the confidence region calculated with error propagation of estimates of PfSR and PfEIR? Likewise, the uncertainty associated with estimates needs to be carefully addressed in all estimates and clearly depicted in the figures.

All confidence regions shown in all plots were obtained using bootstrapping. We have now clarified this.

8. While the online visualisation is good, the manuscript could include static map figures of the results (PfEIR and PfPR). For example, select certain time points with combination of high/low PfEIR and high/low PfPR. This is useful for archive purposes, and also because only the maps in the manuscript can be said to have been peer-reviewed.

We have selected three time points representing high, low and medium transmission and have included static maps thereof in the paper. We have highlighted the selected time points in the main text.[Editors' note: further revisions were suggested prior to acceptance, as described below.]

The manuscript has been improved but there are some remaining issues that need to be addressed, as outlined below:1. Highlight and clarify in the main text the issue related to potential spurious correlation (statistical artifact) between the modelled PfPR and PfEIR.

We have included a paragraph discussing this issue in the Discussion section.

2. Address in the main text how the inclusion of an over dispersion term will change the results of modelling the relationship between PfPR and PfEIR in the six models.

We have included a paragraph discussing this issue in the Discussion section.

3. Introduce into the main text comparison of the estimation PfEIR vs PfPR relationship with previously published models from the literature and explain the observed differences beyond the differences between the modelling approaches.

We have introduced a paragraph in the Discussion to make this comparison.